# Auxiliary Cross-Modal Representation Learning with Triplet Loss Functions for Online Handwriting Recognition

## Abstract

Cross-modal representation learning learns a shared embedding between two or more modalities to improve performance in a given task compared to using only one of the modalities. Cross-modal representation learning from different data types – such as images and time-series data (e.g., audio or text data) – requires a deep metric learning loss that minimizes the distance between the modality embeddings. In this paper, we propose to use the triplet loss, which uses positive and negative identities to create sample pairs with different labels, for cross-modal representation learning between image and time-series modalities (CMR-IS). By adapting the triplet loss for cross-modal representation learning, higher accuracy in the main (time-series classification) task can be achieved by exploiting additional information of the auxiliary (image classification) task. Our experiments on synthetic data and handwriting recognition data from sensor-enhanced pens show improved classification accuracy, faster convergence, and better generalizability.

## 1 Introduction

Cross-modal retrieval (CMR) such as cross-modal representation learning (Peng et al., 2017) for learning across two or more modalities (i.e., image, audio, text and 3D data) has recently garnered substantial interest from the machine learning community. CMR can be applied in a wide range of applications, such as multimedia management (Lee et al., 2020) and identification (Sarafianos et al., 2019). Extracting information from several modalities and adapting the domain with cross-modal learning allows using the information in all domains (Ranjan et al., 2015). Cross-modal representation learning, however, remains challenging due to the *heterogeneity gap* (i.e., inconsistent representation forms of different modalities) (Huang et al., 2020).

A limitation of cross-modal representation learning is that many approaches require the availability of all modalities at inference time. Image-to-caption CMR methods solve this via a separate encoder (Faghri et al., 2018; Chen et al., 2022b). However, in many applications, certain data sources are only available during training by means of elaborate laboratory setups (Lim et al., 2019). For instance, consider a human pose estimation task that uses inertial sensors together with color videos during training, where a camera setup might not be available at inference time due to bad lighting conditions or other application-specific restrictions. Here, a model that allows inference on only the main modality is required, while auxiliary modalities may only be used to improve the training process (as they are not available at inference time) (Hafner et al., 2022). *Learning using privileged information* (Vapnik & Izmailov, 2015) is one approach in the literature that describes and tackles this problem. During training, in addition to $X$, it is assumed that additional *privileged information* $X^*$ is available. However, this *privileged information* is not present in the inference stage (Momeni & Tatwawadi, 2018).

For cross-modal representation learning, we need a deep metric learning technique that aims to transform training samples into feature embeddings that are close for samples that belong to the same class and far apart for samples from different classes (Wei et al., 2016). As deep metric learning requires no model update (simply fine-tuning for training samples of new classes), deep metric learning is an often applied approach for continual learning (Do et al., 2019). Typical deep metric learning methods use not only simple distances (e.g., Euclidean distance), but also highly complex distances (e.g., canonical correlation analysis (Ranjan et al., 2015) and maximum mean discrepancy (Long

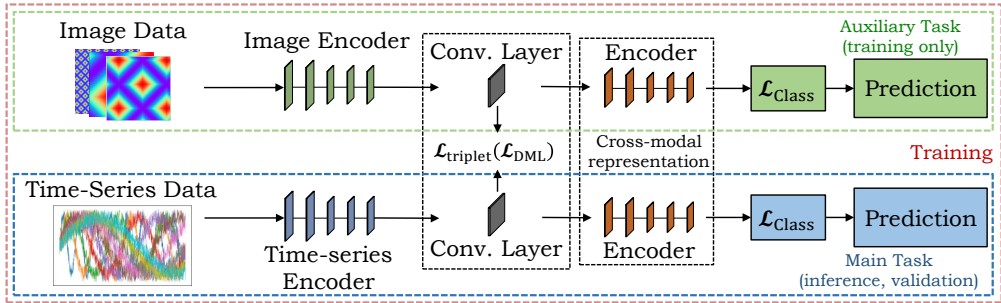

Figure 1: **Method overview:** Cross-modal representation learning between image and time-series data using the triplet loss based on metric learning functions to improve the time-series classification task.

et al., 2015)). While cross-modal representation learning learns representations from all modalities, single-modal learning commonly uses pair-wise learning. The triplet loss (Schroff et al., 2015) selects a positive and negative triplet pair for a corresponding anchor and forces the positive pair distance to be smaller than the negative pair distance. While research of triplet selection for single-modal classification is very advanced (Do et al., 2019; Ott et al., 2022a; Deldari et al., 2022b; Jain et al., 2022; Venkataramanan et al., 2022; Hafner et al., 2022; Wan & Zou, 2021; Li et al., 2021; Kim et al., 2020), pair-wise selection for cross-modal representation learning has mainly been investigated for specific applications (Zhen et al., 2015; Lee et al., 2020; Zhang & Zheng, 2020), i.e., visual semantic embeddings (Chen et al., 2022b; Biten et al., 2022; Diao et al., 2021; Radford et al., 2021).

One exemplary application for cross-modal learning is handwriting recognition (HWR), which can be categorized into offline and online HWR. Offline HWR – such as optical character recognition (OCR) – concerns only analysis of the visual representation of handwriting and cannot be applied for real-time recognition applications (Fahmy, 2010). In contrast, online HWR works on different types of spatio-temporal signals and can make use of temporal information, such as writing speed and direction (Plamondon & Srihari, 2000). As an established real-world application of online HWR, many recording systems make use of a stylus pen together with a touch screen surface (Alimoglu & Alpaydin, 1997). There also exist prototypical systems for online HWR when writing on paper (Chen et al., 2021b; Schrapel et al., 2018; Wang et al., 2013; Deselaers et al., 2015), but these are not yet suitable for real-world applications. However, a novel sensor-enhanced pen based on inertial measurement units (IMUs) may enable new online HWR applications for writing on normal paper. This pen has previously been used for single character (Ott et al., 2020; 2022a;b; Klaß et al., 2022) and sequence (Ott et al., 2022c) classification. However, the accuracy of previous online HWR methods is limited, due to the following reasons: (1) The size of datasets is limited, as recording larger amounts of data is time-consuming. (2) Extracting important spatio-temporal features is important. (3) Training a writer-independent classifier is challenging, as different writers can have notably different writing styles. (4) Evaluation performance drops for under-represented groups, i.e., left-handed writers. (5) The model overfits to seen words that can be addressed with generated models. A possible solution is to combine datasets of different modalities using cross-modal representation learning to increase generalizability. In this work, we combine offline HWR from generated images (i.e., OCR) and online HWR from sensor-enhanced pens by learning a common representation between both modalities. The aim is to integrate information on OCR – i.e., typeface, cursive or printed writing, and font thickness – into the online HWR task – i.e., writing speed and direction (Vinciarelli & Perrone, 2003).

**Our Contribution.** Models that use rich data (e.g., images) usually outperform those that use a less rich modality (e.g., time-series). We therefore propose to train a shared representation using the triplet loss between pairs of image and time-series data to learn a cross-modal representation between both modality embeddings (cf. Figure 1). This allows for improving the accuracy of single-modal inference in the main task. Cross-modal learning between images and time-series data is rare. Furthermore, we propose a novel dynamic margin for the triplet loss based on the Edit distance. We prove the efficacy of our metric learning-based triplet loss for cross-modal representation learning both with simulated data and in a real-world application. More specifically, our proposed cross-modal representation learning technique 1) improves the multivariate time-series classification accuracy and convergence, 2) results in a small time-series-only network independent from the image modality while allowing for fast inference, and 3) has better generalizability and adaptability (Huang et al., 2020). Our approach shows that the recent methods ScrabbleGAN (Fogel et al., 2020) and OrigamiNet (Yousef & Bishop, 2020) are applicable in the real-world setup of offline HWR to

enhance the online HWR task. We provide an extensive overview and technical comparison of related methods. Code and datasets are available upon publication.[1]

The paper is organized as follows. Section 2 discusses related work followed by the mathematical foundation of our method in Section 3. The methodology is described in Section 4 and the results are discussed in Section 5.

## 2 Related Work

In this section, we discuss related work – particularly, methods of offline HWR (in Section 2.1) and online HWR (in Section 2.2). We summarize approaches for learning a cross-modal representation from different modalities (in Section 2.3), pairwise and triplet learning (in Section 2.4), and deep metric learning (in Section 2.5) to minimize the distance between feature embeddings.

### 2.1 Offline Handwriting Recognition

In the following, we give a brief overview of offline HWR methods to select a suitable lexicon and language model-free method. For an overview of offline and online HWR datasets, see (Plamondon & Srihari, 2000; Hussain et al., 2015). Table 1 presents related work. For a more detailed overview, see Table 9 in the Appendix A.1. In Table 1, we refer to the use of a language model as LM with $k$ and identify the data level on which the method works – i.e., paragraph or full-text level, line level, and word level. We present evaluation results for the IAM-OffDB (Liwicki & Bunke, 2005) and RIMES (Grosicki & El-Abed, 2011) datasets. We show the character error rate (CER) – the percentage of characters that were incorrectly predicted (the lower, the better) – and the word error rate (WER) – a common performance metric on word level instead of the phoneme level (the lower, the better); for further information, see Section 5.2.

Methods for offline HWR range from hidden Markov models (HMMs) – such as (Bertolami & Bunke, 2018; Dreuw et al., 2011; Li et al., 2014; Pastor-Pellicer et al., 2015; España-Boquera et al., 2011; Kozielski et al., 2013; Doetsch et al., 2014) – to deep learning techniques that became predominant in 2014, such as convolutional neural networks (CNNs) as the methods by Poznanski & Wolf (2016); Wang et al. (2020b). The gated text recognizer (Yousef et al., 2020) aims to automate the feature extraction from raw input signals with a minimum required domain knowledge. The fully convolutional network without recurrent connections is trained with the CTC loss. Thus, the gated text recognizer module can handle arbitrary input sizes and can recognize strings with arbitrary lengths. This module has been used for OrigamiNet (Yousef & Bishop, 2020) which is a segmentation-free multi-line or full-page recognition system. OrigamiNet yields state-of-the-art results on the IAM-OffDB dataset, and shows improved performance of gated text recognizer over VGG and ResNet26. Hence, we use the gated text recognizer module as our visual feature encoder for offline HWR. Furthermore, temporal convolutional networks (TCNs) employ the temporal context of the handwriting – such as the methods (Sharma et al., 2020; Sharma & Jayagopi, 2021). More prominent became recurrent neural networks (RNNs) including long short-term memories (LSTMs), bidirectional LSTMs (BiLSTMs) (Chowdhury & Vig, 2018; Sueiras et al., 2018; Ingle et al., 2019; Michael et al., 2019), and multidimensional RNNs (MDRNN, MDLSTM) (Graves & Schmidhuber, 2008; Bluche, 2016; Voigtlaender et al., 2016; Chen et al., 2017; Bluche et al., 2017; Castro et al., 2018; Krishnan et al., 2018). Sequential architectures are perfect to fit text lines, due to the probability distributions over sequences of characters, and due to the inherent temporal aspect of text (Kang et al., 2022). Graves et al. (2009) introduced the BiLSTM layer in combination with the CTC loss. Pham et al. (2014) showed that the performance of LSTMs can be greatly improved using dropout. GCRNN (Bluche & Messina, 2017) combines a convolutional encoder (aiming for generic and multilingual features) and a BiLSTM decoder predicting character sequences. Additionally, Puigcerver (2017) proposed a CNN+BiLSTM architecture (CNN-1DLSTM-CTC) that uses the CTC loss. Further methods that combine CNNs with RNNs are (Liang et al., 2017; Sudholt & Fink, 2018; Xiao & Cho, 2016), while BiLSTMs are utilized in (Carbune et al., 2020; Tian et al., 2019). Recent methods are generative adversarial networks (GANs) and Transformers.

The first approach by Graves (2014) was a method to synthesize online data based on RNNs. The technique HWGAN by Ji & Chen (2020) extends this method by adding a discriminator $\mathcal{D}$. DeepWriting (Aksan et al., 2018) is a GAN that is capable of disentangling style from content and thus making digital ink editable. Haines et al. (2016) proposed a method to generate handwriting based on a specific author with learned parameters for spacing, pressure, and line

---

[1]Code and datasets: https://www.anonymous-submission.org *(will be updated after acceptance)*

Table 1: Evaluation results (WER and CER in %) of different methods on the IAM-OffDB (Liwicki & Bunke, 2005) and RIMES (Grosicki & El-Abed, 2011) datasets. The table is sorted by year.

| | Method | Information | LM size $k$ | P | L | W | IAM-OffDB WER | CER | RIMES WER | CER |
|---|---|---|---|---|---|---|---|---|---|---|
| **RNN** | Michael et al. (2019) | Seq2seq CNN+BiLSTM (64 × width) | | | × | | - | 5.24 | - | - |
| | FPN (Carbonell et al., 2019) | Feature Pyramid Network, 150 dpi | | | × | | - | 15.60 | - | - |
| | AFDM (Bhunia et al., 2019) | AFD module | w/ | | | | 8.87 | 5.94 | 6.31 | 3.17 |
| **CNN** | Poznanski & Wolf (2016) | CNN + connected branches, CCA | w/ | | | | 6.45 | 3.44 | 3.90 | 1.90 |
| | Gated text recognizer (Yousef et al., 2020) | CNN+CTC (32 × width) | w/o | | × | | - | 4.90 | - | - |
| | OrigamiNet (Yousef & Bishop, 2020) | VGG (500 × 500) | × | × | | | - | 51.37 | - | - |
| | | VGG (500 × 500), w/o LN | w/o | × | × | | - | 34.55 | - | - |
| | | ResNet26 (500 × 500), w/o LN | w/o | × | × | | - | 10.03 | - | - |
| | | ResNet26 (500 × 500), w/ LN | w/o | × | × | | - | 7.24 | - | - |
| | | ResNet26 (500 × 500), w/o LN | w/o | × | × | | - | 8.93 | - | - |
| | | ResNet26 (500 × 500), w/ LN | w/o | × | × | | - | 6.37 | - | - |
| | | ResNet26 (500 × 500), w/o LN | w/o | × | × | | - | 76.90 | - | - |
| | | ResNet26 (500 × 500), w/ LN | w/o | × | × | | - | 6.13 | - | - |
| | | GTR-8 (500 × 500), w/o LN | w/o | × | × | | - | 72.40 | - | - |
| | | GTR-8 (500 × 500), w/ LN | w/o | × | × | | - | 5.64 | - | - |
| | | GTR-8 (750 × 750), w/ LN | w/o | × | × | | - | 5.50 | - | - |
| | | GTR-12 (750 × 750), w/ LN | w/o | × | × | | - | 4.70 | - | - |
| | DAN (Wang et al., 2020b) | Decoupled attention module | w/o | | × | | 19.60 | 6.40 | 8.90 | 2.70 |
| **GAN** | ScrabbleGAN (Fogel et al., 2020) | Original data | w/o | | | | 25.10 | - | 12.29 | - |
| | | Augmentation | w/o | | | | 24.73 | - | 12.24 | - |
| | | Augmentation + 100k synth. | w/o | | | | 23.98 | - | 11.68 | - |
| | | Augmentation + 100k synth. + Refine | w/o | | | | 23.61 | - | 11.32 | - |
| **Transformer** | Kang et al. (2022) | Self-attention for text/images | w/o | | × | | 15.45 | 4.67 | - | - |
| | FPHR (Singh & Karayev, 2021) | CNN encoder, Transformer decoder | w/o | × | | | - | 6.70 | - | - |
| | | With augmentation | w/o | × | | | - | 6.30 | - | - |

**Abbreviations.** Size $k$ of the language model (LM), i.e., with (w/) or without (w/o) a LM. P: paragraph or full text level, L: line level, LN: layer normalization, CER: character error rate, WER: word error rate, GTR: gated text recognizer, seq2seq: sequence-to-sequence, GAN: generative adversarial network, CTC: connectionist temporal classification

thickness. Alonso et al. (2019) used a BiLSTM to obtain an embedding of the word to be rendered and added an auxiliary network as a recognizer $\mathcal{R}$. The model is trained with a combination of an adversarial loss and the CTC loss. ScrabbleGAN by Fogel et al. (2020) is a semi-supervised approach that can arbitrarily generate many images of words with arbitrary length from a generator $\mathcal{G}$ to augment handwriting data and uses a discriminator $\mathcal{D}$ and recognizer $\mathcal{R}$. The paper proposes results for original data with random affine augmentation using synthetic images and refinement.

RNNs prevent parallelization, due to their sequential pipelines. Kang et al. (2022) introduced a non-recurrent model by the use of Transformer models with multi-head self-attention layers at the textual and visual stages. Their method works for any pre-defined vocabulary. For the feature encoder, they used modified ResNet50 models. The full page HTR (FPHR) method by Singh & Karayev (2021) uses a CNN as an encoder and a Transformer as a decoder with positional encoding.

## 2.2 Online Handwriting Recognition

Motion-based handwriting (Chen et al., 2021b) and air-writing (Zhang et al., 2022) from sensor-enhanced devices have been extensively investigated. While such motions are spacious, the hand and pen motions for writing on paper are comparatively small-scale (Bu et al., 2021). Research for classifying text from sensor-enhanced pens has recently attracted substantial interest. He et al. (2022a) use acceleration and audio data of handwritten actions for character recognition. Furthermore, recent publications came up with similar developments that are only prototypical, for example, Singh & Chaturvedi (2023); He et al. (2022b); Alemayoh et al. (2022). Hence, there is already a lot of interest and future technical advancements will further boost this. The novel sensor-enhanced pen based on IMUs (Ott et al., 2020) enables new applications for writing on paper. Note that this pen is a finished product and can be bought. Data collection and processing is straightforward and allows applications to be easy to implement in real-world. Ott et al. (2020) published the OnHW-chars dataset containing single characters. Klaß et al. (2022) evaluated the aleatoric and epistemic uncertainty to show the domain shift between right- and left-handed writers. Ott et al. (2022a) reduced this domain shift by adapting feature embeddings based on transformations from optimal transport techniques. Kreß et al. (2022) presented an approach for distributing the computational workload between a sensor pen and a mobile device (i.e., smartphone or tablet) for handwriting recognition, as interference on mobile devices leads to high system requirements. Ott et al. (2022b) reconstructed the trajectory of the pen tip for single characters written on tablets

from IMU data and cameras pointing at the pen tip. A more challenging task than single-character classification is the classification of sequences (i.e., words or equations). Ott et al. (2022c) proposed several sequence-based datasets and a large benchmark of convolutional, recurrent, and Transformer-based architectures, loss functions, and augmentation techniques. While Wegmeth et al. (2021) combined a binary random forest to classify the writing activity and a CNN for windows of single-label predictions, Bronkhorst (2021) highlighted the effectiveness of Transformers for classifying equations. Methods such as (Singh & Karayev, 2021) cannot be applied to this online task, as these methods are designed for image-based (offline) HWR, and traditional methods such as (Carbune et al., 2020) for online HWR are based on online trajectories written on tablets. Recently, Azimi et al. (2022) evaluated further machine and deep learning models as well as deep ensembles on the single OnHW-chars dataset.

## 2.3 Cross-Modal Representation Learning

For traditional methods that learn a cross-modal representation, a cross-modal similarity for the retrieval can be calculated with linear projections (Rasiwasia et al., 2010). However, cross-modal correlation is highly complex, and hence, recent methods are based on a *modal-sharing network* to jointly transfer non-linear knowledge from a single modality to all modalities (Wei et al., 2016). Huang et al. (2020) use a *cross-modal network* between different modalities (image to video, text, audio and 3D models) and a *single-modal network* (shared features between images of source and target domains). They use two convolutional layers (similar to our proposed architecture) that allow the model to adapt by using more trainable parameters. However, while their auxiliary network uses the same modality, the auxiliary network of the proposed method in this paper is based on another modality. Lee et al. (2020) learn a cross-modal embedding between video frames and audio signals with graph clusters, but both modalities must be available at inference. Sarafianos et al. (2019) proposed an image-text modality adversarial matching approach that learns modality-invariant feature representations, but their projection loss is only used for learning discriminative image-text embeddings. Hafner et al. (2022) propose a model for single-modal inference. However, they use image and depth modalities for person re-identification without a time-series component, which makes the problem considerably different. Lim et al. (2019) handled multi-sensory modalities for 3D models only. For an overview of CMR, see Deldari et al. (2022a). An overview of relevant CMR methods is given in Table 2, and more detailed in the Appendix A.2. With respect to the kind of the modality, the work by Gu et al. (2022); Ott et al. (2022a) is closest, while the applications in (Ott et al., 2022a; Zeng et al., 2017; Wan & Zou, 2021) of handwriting recognition are relevant.

## 2.4 Pairwise and Triplet Learning

Networks trained for a classification task can produce useful feature embeddings with efficient runtime complexity $\mathcal{O}(NC)$ per epoch, where $N$ is the number of training samples and $C$ is the number of classes. However, the classical cross-entropy (CE) loss is only partly useful for deep metric learning, as it ignores how close each point is to its class centroid (or how far apart each point is from other class centroids). CE variations (e.g., for face recognition) that learn angularly discriminative features have also been developed (Liu et al., 2017). The *pairwise contrastive loss* (Chopra et al., 2005) minimizes the distance between feature embedding pairs of the same class and maximizes the distance between feature embedding pairs of different classes depending on a margin parameter. The drawback is that the optimization of positive pairs is independent of negative pairs, but the optimization should force the distance between positive pairs to be smaller than negative pairs (Do et al., 2019).

The *triplet loss* (Yoshida et al., 2019) addresses this by defining an anchor and a positive point as well as a negative point and forces the positive pair distance to be smaller than the negative pair distance by a certain margin. The runtime complexity of the triplet loss is $\mathcal{O}(N^3/C)$ and can be computationally challenging for large training sets. Hence, several approaches exist to reduce this complexity, such as hard or semi-hard triplet mining (Schroff et al., 2015) and smart triplet mining (Harwood et al., 2017). Often, data evolve over time, and hence, Semedo & Magalhães (2020) proposed a formulation of the triplet loss where the traditional static *margin* is superseded by a temporally adaptive maximum margin function. While Zeng et al. (2017); Li et al. (2021) combine the triplet loss with the CE loss, Guo et al. (2019) use a triplet selection with $L_2$-normalization for language modeling, but considered all negative pairs for triplet selection with fixed similarity intensity parameter. The proposed method uses a triplet loss with a dynamic margin together with a novel word-level triplet selection. The TNN-C-CCA (Zeng et al., 2020) also uses the triplet loss on embeddings between an anchor from audio data and positive and negative samples from visual data and the cosine similarity for the final representation comparison. In image-to-caption CMR tasks, the most common design is separated encoders that allow the separated inference without the other modality (Faghri et al., 2018; Chen et al.,

Table 2: Overview of cross-modal and pairwise learning techniques using the modalities video (V), images (I), audio (A), text (T), sensors (S), or haptic (H). Data from sensors are represented by time-series from inertial, biological, or environmental sensors. We indicate cross-modal learning from the same modality with $\times^n$ with $n$ modalities. If $n$ is unspecified, the method can potentially work with an arbitrary number of modalities.

| Method (sorted by year) | V | I | A | T | S | H | Pairwise Learning | Deep Metric Learning Loss/Objective | Application |
|---|---|---|---|---|---|---|---|---|---|
| OxfordNet (Kiros et al., 2014) | | $\times$ | | $\times$ | | | Contrastive | Cosine similarity | Visual semantic embedding |
| DAN (Long et al., 2015) | | $\times^2$ | | | | | Pairwise | Kernelized MMD | Domain adaptation |
| FaceNet (Schroff et al., 2015) | | $\times$ | | | | | Triplet | Euclidean semantic matching | Face recognition, clustering various recognition tasks |
| TristouNet (Bredin, 2017) | | | $\times$ | | | | Triplet | Euclidean | Speech classification |
| Zeng et al. (2017) | | $\times$ | | | | | Triplet | CE, conditional random field | Handwritten Chinese characters recognition |
| VSE++ (Faghri et al., 2018) | | $\times$ | | $\times$ | | | Triplet | Similarity: inner product | Visual semantic embedding |
| PIE-Nets (Song & Soleymani, 2019) | $\times$ | $\times$ | | $\times$ | | | Pairwise | Diversity, MIL, MMD | Visual semantic embedding |
| Zhang et al. (2019) | | $\times$ | | | | | STriplet+triplet | Cosine similarity | Relationship understanding |
| GMN (Lim et al., 2019) | | $\times^n$ | $\times^n$ | | | $\times^n$ | Pairwise | Cross-modal generation | Multisensory 3D scenes |
| CTM (Guo et al., 2019) | $\times$ | | | $\times$ | | | Triplet | CTC, CE , $L_2$ correlation | Sentence translation |
| ActiveSet+RRPB (Yoshida et al., 2019) | | $\times$ | | | | | Smart triplet | Semidefinite constraint | General |
| CrossATNet (Chaudhuri et al., 2020) | | $\times^2$ | | $\times$ | | | Triplet | MSE | Zero-shot learning, sketches |
| MHTN (Huang et al., 2020) | $\times$ | $\times$ | $\times$ | $\times$ | | | Pairwise, contr. | MMD, Euclidean | CMR |
| Proxy-Anchor (Kim et al., 2020) | | $\times$ | | | | | Pair+proxy | Cosine similarity | Image classification |
| CLIP (Radford et al., 2021) | | $\times$ | | $\times$ | | | Contrastive | CE, Cosine similarity | OCR, action/object recognition |
| MCN (Chen et al., 2021a) | $\times$ | | $\times$ | $\times$ | | | Contrastive | Similarity, reconstruction | Multimodal clustering |
| VATT (Akbari et al., 2021) | $\times$ | | $\times$ | $\times$ | | | Contrastive | CC, NCE, MIL-NCE | Transformer for CMR |
| Wan & Zou (2021) | | $\times$ | | | | | Dual triplet | Euclidean | Signature verification |
| Wang et al. (2021) | $\times$ | | $\times^2$ | | | | Contrastive | Cosine similarity | Audio classification |
| Hafner et al. (2022) | | $\times^2$ | | | | | Triplet | Softmax, MSE | Person re-identification |
| AlignMixup (Venkataramanan et al., 2022) | | $\times^2$ | | | | | Pairwise | Sinkhorn transport | Data augmentation for interpolation |
| VSE$_\infty$ (Chen et al., 2022b) | | $\times$ | | $\times$ | | | Triplet | Similarity | Visual semantic embedding |
| AudioCLIP (Guzhov et al., 2022) | | $\times$ | $\times$ | $\times$ | | | Contrastive | Cosine similarity, symmetric CE | Environmental sound classification |
| ColloSSL (Jain et al., 2022) | | | | | $\times^n$ | | Contrastive | CE, Cosine similarity | Human-activity recognition |
| COCOA (Deldari et al., 2022b) | | | | | $\times^n$ | | Contrastive | Cosine similarity | General time-series |
| MM-ALT (Gu et al., 2022) | $\times$ | | $\times$ | $\times$ | | | Pairwise | CTC, residual attention | Lyric transcription |
| FLAVA (Singh et al., 2022) | | $\times$ | | $\times$ | | | Contrastive | Cosine similarity temperature scaling | Visual semantic embedding |
| ConceptBeam (Ohishi et al., 2022) | | $\times$ | $\times^n$ | | | | Triplet | Cosine similarity | Target speech extraction |
| Falcon et al. (2022) | $\times$ | | | $\times$ | | | Contrastive | - | Text-video retrieval, kitchen |
| C$^3$CMR (Wang et al., 2022) | | $\times$ | | $\times$ | | | Triplet | CE, cosine similarity | Visual semantic embedding |
| Ott et al. (2022a) | | | | | $\times^2$ | | Classwise | CE, HoMM/CC/PC | Online HWR |
| **CMR-IS (Ours, 2022)** | | $\times$ | | $\times$ | | | Contr., triplet | CTC, MSE/CC/PC/KL | Online HWR |

**Abbreviations.** CE: cross-entropy, CTC: connectionist temporal classification, MSE: mean squared error, CC: cross-correlation, PC: Pearson correlation, MMD: maximum mean discrepancy, HoMM: higher-order moment matching, CCA: canonical correlation analysis, MIL: multiple-instance learning, MI: mutual information, NCE: noise contrastive estimation, VSE: visual semantic embedding

2022b). We choose a similar separate cross-modal encoder for single-modal inference. CrossATNet (Chaudhuri et al., 2020), another triplet loss-based method that uses single class labels, defines class sketch instances as the anchor, the same class image instance as the positive sample, and a different class image instance as the negative sample. While the previous methods are based on a triplet selection method using single-label classification, related work exists for using the triplet loss for sequence-based classification (i.e., from texts) (Gordo & Larlus, 2017; Deng et al., 2018; Zhang et al., 2019; Bredin, 2017). To the best of our knowledge, no approach so far has used triplet-based cross-modal learning based on the Edit distance between words. Most relevant are the works by Wang et al. (2022); Ohishi et al. (2022); Chen et al. (2022b); Hafner et al. (2022); Chaudhuri et al. (2020) that use the triplet loss, but without a dynamic margin.

## 2.5 Deep Metric Learning

As deep metric learning is a very broad and advanced field, only the most related work is described here. For an overview of deep metric learning, see Musgrave et al. (2020). Most of the related work uses the Euclidean metric as distance loss, although the triplet loss can be defined based on any other (sub-)differentiable distance metric. Wan & Zou (2021) proposed a method for offline signature verification based on a dual triplet loss that uses the Euclidean

space to project an input image to an embedding function. While Rantzsch et al. (2016) use the Euclidean metric to learn the distance between feature embeddings, Zeng et al. (2017) use the Cosine similarity. Hermans et al. (2017) state that using the *non-squared* Euclidean distance is more stable, while the *squared* distance made the optimization more prone to collapsing. Recent methods extend the canonical correlation analysis (CCA) (Ranjan et al., 2015) that learns linear projection matrices by maximizing pairwise correlation of cross-modal data. To share information between the same modality (i.e., images), the maximum mean discrepancy (MMD) (Long et al., 2015) is typically minimized.

## 3 Methodological Background

We define the problem of cross-mdoal representation learning and present deep metric learning loss functions in Section 3.1. In Section 3.2, we propose the triplet loss for cross-modal learning.

### 3.1 Cross-Modal Retrieval for Time-Series and Image Classification

A multivariate time-series $\mathbf{U} = \{\mathbf{u}_1, \ldots, \mathbf{u}_m\} \in \mathbb{R}^{m \times l}$ is an ordered sequence of $l \in \mathbb{N}$ streams with $\mathbf{u}_i = (u_{i,1}, \ldots, u_{i,l}), i \in \{1, \ldots, m\}$, where $m \in \mathbb{N}$ is the length of the time-series. The multivariate time-series training set is a subset of the array $\mathcal{U} = \{\mathbf{U}_1, \ldots, \mathbf{U}_{n_U}\} \in \mathbb{R}^{n_U \times m \times l}$, where $n_U$ is the number of time-series. Let $\mathbf{X} \in \mathbb{R}^{h \times w}$ with entries $x_{i,j} \in [0, 255]$ represent an image from the image training set. The image training set is a subset of the array $\mathcal{X} = \{\mathbf{X}_1, \ldots, \mathbf{X}_{n_X}\} \in \mathbb{R}^{n_X \times h \times w}$, where $n_X$ is the number of time-series. The aim of joint multivariate time-series and image classification tasks is to predict an unknown class label $y \in \Omega$ for single class prediction or $\mathbf{y} \in \Omega$ for sequence prediction for a given multivariate time-series or image (see also Section 4.2). The time-series samples denote the main training data, while the image samples represent the privileged information that is not used for inference. In addition to good prediction performance, the goal is to learn representative embeddings $f_c(\mathbf{U})$ and $f_c(\mathbf{X}) \in \mathbb{R}^{q \times t}$ to map multivariate time-series and image data into a feature space $\mathbb{R}^{q \times t}$, where $f_c$ is the output of the convolutional layer(s) $c \in \mathbb{N}$ of the latent representation and $q \times t$ is the dimension of the layer output.

We force the embedding to live on the $q \times t$-dimensional hypersphere by using $\texttt{softmax}$ – i.e., $||f_c(\mathbf{U})||_2 = 1$ and $||f_c(\mathbf{X})||_2 = 1 \forall c$ (see (Weinberger et al., 2005)). In order to obtain a small distance between the embeddings $f_c(\mathbf{U})$ and $f_c(\mathbf{X})$, we minimize deep metric learning functions $\mathcal{L}_{\text{DML}}(f_c(\mathbf{X}), f_c(\mathbf{U}))$. Well-known deep learning metric are the distance-based mean squared error (MSE) $\mathcal{L}_{\text{MSE}}$, the spatio-temporal cosine similarity (CS) $\mathcal{L}_{\text{CS}}$, the Pearson correlation (PC) $\mathcal{L}_{\text{PC}}$, and the distribution-based Kullback-Leibler (KL) divergence $\mathcal{L}_{\text{KL}}$. In our experiments, we additionally evaluate the kernalized maximum mean discrepancy (kMMD) $\mathcal{L}_{\text{kMMD}}$, Bray Curtis (BC) $\mathcal{L}_{\text{BC}}$, and Poisson $\mathcal{L}_{\text{PO}}$ losses. We study their performance in Section 5. A combination of classification and cross-modal representation learning losses can be realized by dynamic weight averaging (Liu et al., 2019) as a multi-task learning approach that performs dynamic task weighting over time (see Appendix A.3).

### 3.2 Contrastive Learning and Triplet Loss

While the training with the previous loss functions uses inputs where the image and multivariate time-series have the same label, pairs with similar but different labels can improve the training process. This can be achieved using the triplet loss (Schroff et al., 2015), which enforces a margin between pairs of image and multivariate time-series data with the same identity to all other different identities. As a consequence, the convolutional output for one and the same label lives on a manifold, while still enforcing the distance – and thus, discriminability – to other identities.

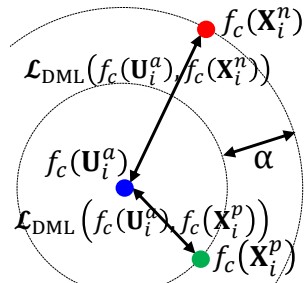

Figure 2: Triplet pair.

Therefore, we seek to ensure that the embedding of the multivariate time-series $\mathbf{U}_i^a$ (*anchor*) of a specific label is closer to the embedding of the image $\mathbf{X}_i^p$ (*positive*) of the same label than it is to the embedding of any image $\mathbf{X}_i^n$ (*negative*) of another label (see Figure 2). Thus, we want the following inequality to hold for all training samples $\left(f_c(\mathbf{U}_i^a), f_c(\mathbf{X}_i^p), f_c(\mathbf{X}_i^n)\right) \in \Phi$:

$$\mathcal{L}_{\text{DML}}\left(f_c(\mathbf{U}_i^a), f_c(\mathbf{X}_i^p)\right) + \alpha < \mathcal{L}_{\text{DML}}\left(f_c(\mathbf{U}_i^a), f_c(\mathbf{X}_i^n)\right), \tag{1}$$

where $\mathcal{L}_{\text{DML}}\left(f_c(\mathbf{X}), f_c(\mathbf{U})\right)$ is a deep metric learning loss, $\alpha$ is a margin between positive and negative pairs, and $\Phi$ is the set of all possible triplets in the training set. The *contrastive loss* minimizes the distance of the anchor to the

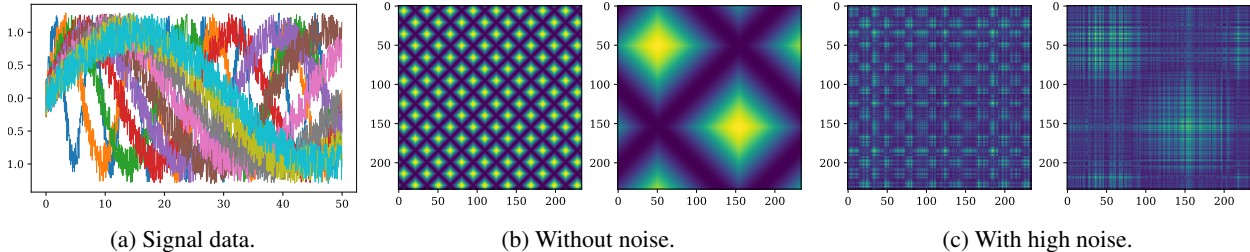

(a) Signal data.        (b) Without noise.        (c) With high noise.

Figure 3: Synthetic signal data (a) for 10 classes, and image data (b-c) for classes 0 (left) and 6 (right).

positive sample and separately maximizes the distance to the negative sample. Instead, based on (1), we can formulate a differentiable loss function - the *triplet loss* - that we can use for optimization:

$$\mathcal{L}_{\text{trpl},c}(\mathbf{U}^a, \mathbf{X}^p, \mathbf{X}^n) = \sum_{i=1}^{N} \max \left[ \mathcal{L}_{\text{DML}}\big(f_c(\mathbf{U}_i^a), f_c(\mathbf{X}_i^p)\big) - \mathcal{L}_{\text{DML}}\big(f_c(\mathbf{U}_i^a), f_c(\mathbf{X}_i^n)\big) + \alpha, 0 \right], \qquad (2)$$

where $c \in \mathbb{N}$.[2] Selecting negative samples that are too close to the anchor (in relation to the positive sample) can cause slow training convergence. Hence, triplet selection must be handled carefully and with consideration for each specific application (Do et al., 2019). We choose negative samples based on the class distance (single labels) and on the Edit distance (sequence labels) (see Section 4.2).

# 4 Method

We now demonstrate the efficacy of our proposal. In Section 4.1, we generate sinusoidal time-series with introduced noise (main task) and compute the corresponding Gramian angular summation field with different noise parameters (auxiliary task) (see Figure 1). In Section 4.2, we combine online (inertial sensor signals, main task) and offline data (visual representations, auxiliary task) for HWR with sensor-enhanced pens. This task is particularly challenging, due to different data representations based on images and multivariate time-series data. For both applications, our approach allows to only use the main modality (i.e., multivariate time-series) for inference. We further analyze and evaluate different deep metric learning functions to minimize the distance between the learned embeddings.

## 4.1 Cross-Modal Learning on Synthetic Data

We first investigate the influence of the triplet loss for cross-modal learning between synthetic time-series and image-based data as a sanity check. For this, we generate signal data of 1,000 timesteps with different frequencies for 10 classes (see Figure 3a) and add noise from a continuous uniform distribution $U(a, b)$ for $a = 0$ and $b = 0.3$. We use a recurrent CNN with the CE loss to classify these signals. From each signal without noise, we generate a Gramian angular summation field (Wang & Oates, 2015). For classes with high frequencies, this results in a fine-grained pattern, and for low frequencies in a coarse-grained pattern. We generate Gramian angular summation fields with different added noise between $b = 0$ (Figure 3b) and $b = 1.95$ (Figure 3c). A small CNN classifies these images with the CE loss. To combine both networks, we train each signal-image pair with the triplet loss. As the frequency of the sinusoidal signal is closer for more similar class labels, the distance in the manifold embedding should also be closer. For each batch, we select negative sample pairs for samples with the class label $CL = 1 + \lfloor \frac{\max_e - e - 1}{25} \rfloor$ as the lower bound for the current epoch $e$ and the maximum epoch $\max_e$. We set the margin $\alpha$ in the triplet loss separately for each batch such that $\alpha = \beta \cdot (CL_p - CL_n)$ depends on the positive $CL_p$ and negative $CL_n$ class labels of the batch and is in the range $[1, 5]$ with $\beta = 0.1$. The batch size is 100 and $\max_e = 100$. Appendix A.4 provides further details. This combination of the CE loss with the triplet loss can lead to a mutual improvement of the utilization of the classification task and embedding learning.

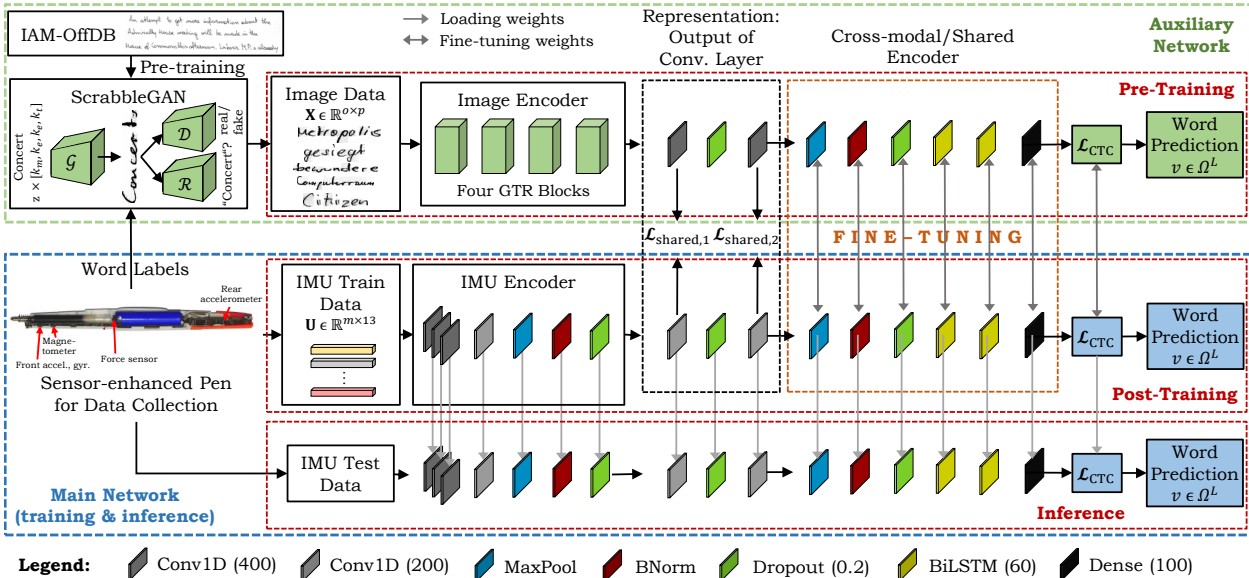

Figure 4: **Detailed method overview:** The middle pipeline consists of data recording with a sensor-enhanced pen, feature extraction of inertial multivariate time-series data, and word classification with CTC. We generate image data with the pre-trained ScrabbleGAN for corresponding word labels. The top pipeline (four gated text recognizer blocks) extracts features from images. The distances of the embeddings are minimized with the triplet loss and deep metric learning functions. The classification network with two BiLSTM layers are fine-tuned for the OnHW task for a cross-modal representation.

## 4.2 Cross-Modal Learning for HWR

**Method Overview.** Figure 4 gives a method overview. The main task is online HWR to classify words written with a sensor-enhanced pen and represented by multivariate time-series of the different pen sensors. To improve the classification task with a better generalizability, the auxiliary network performs offline HWR based on an image input. We pre-train ScrabbleGAN (Fogel et al., 2020) on the IAM-OffDB (Liwicki & Bunke, 2005) dataset. For all time-series word labels, we then generate the corresponding image as the positive time-series-image pair. Each multivariate time-series and each image is associated with $\mathbf{y}$ – a sequence of $L$ class labels from a pre-defined label set $\Omega$ with $K$ classes. For our classification task, $\mathbf{y} \in \Omega^L$ describes words. The multivariate time-series training set is a subset of the array $\mathcal{U}$ with labels $\mathcal{Y}_U = \{\mathbf{y}_1, \ldots, \mathbf{y}_{n_U}\} \in \Omega^{n_U \times L}$. The image training set is a subset of the array $\mathcal{X}$, and the corresponding labels are $\mathcal{Y}_X = \{\mathbf{y}_1, \ldots, \mathbf{y}_{n_X}\} \in \Omega^{n_X \times L}$. Offline HWR techniques are based on Inception, ResNet34, or gated text recognizer (Yousef et al., 2020) modules. The architecture of the online HWR method consists of an IMU encoder with three 1D convolutional layers of size 400, a convolutional layer of size 200, a max pooling and batch normalization, and a dropout of 20%. The online method is improved by sharing layers with a common representation by minimizing the distance of the feature embedding of the convolutional layers $c \in \{1, 2\}$ (integrated in both networks) with a shared loss $\mathcal{L}_{\text{shared},c}$. We set the embedding size $\mathbb{R}^{q \times t}$ to $400 \times 200$. Both networks are trained with the connectionist temporal classification (CTC) (Graves et al., 2009) loss $\mathcal{L}_{\text{CTC}}$ to avoid pre-segmentation of the training samples by transforming the network outputs into a conditional probability distribution over label sequences.

**Datasets for Online HWR.** We make use of two word datasets proposed in (Ott et al., 2022c). These datasets are recorded with a sensor-enhanced pen that uses two accelerometers (3 axes each), one gyroscope (3 axes), one magnetometer (3 axes), and one force sensor at 100 Hz (Ott et al., 2020; 2022b). One sample of size $m \times l$ represents an multivariate time-series of a written word of $m$ timesteps from $l = 13$ sensor channels. One word is a sequence of small or capital characters (52 classes) or with mutated vowels (59 classes). The *OnHW-words500* dataset contains 25,218 samples where each of the 53 writers contributed the same 500 words. The *OnHW-wordsRandom* dataset

---

[2]To have a larger number of trainable parameters in the latent representation with a greater depth, we evaluate one and two stacked convolutional layers, each trained with a shared loss $\mathcal{L}_{\text{trpl},c}$.

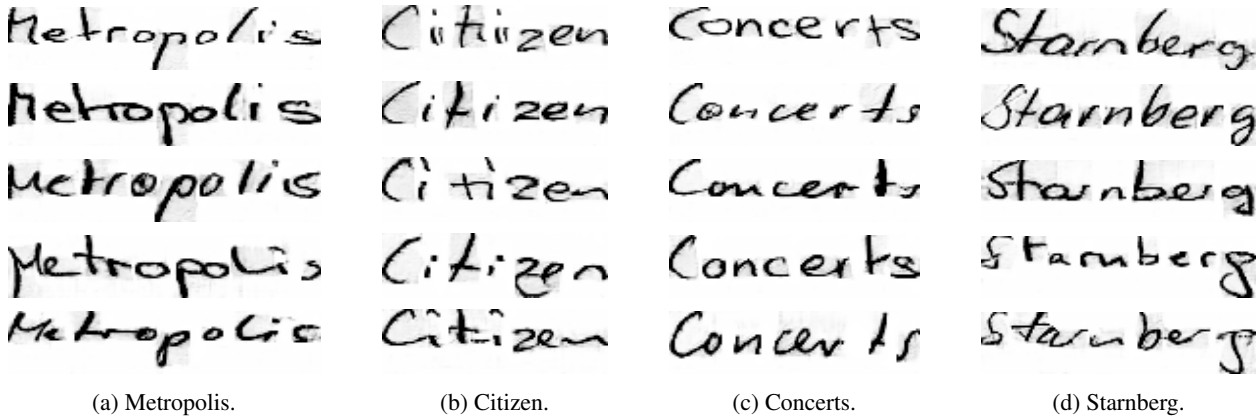

(a) Metropolis.          (b) Citizen.          (c) Concerts.          (d) Starnberg.

Figure 6: Overview of four generated words with ScrabbleGAN (Fogel et al., 2020) with various text styles.

contains 14,641 randomly selected words from 54 writers. For both datasets, 80/20 train/validation splits are available for writer-(in)dependent (WD/WI) tasks. We transform (zero padding, interpolation) all samples to 800 timesteps. For more information on the datasets, see Ott et al. (2022c).

**Image Generation for Offline HWR.** In order to couple the online time-series data with offline image data, we use a generative adversarial network (GAN) to arbitrarily generate many images. ScrabbleGAN (Fogel et al., 2020) is a state-of-the-art semi-supervised approach that consists of a generator $\mathcal{G}$ that generates images of words with arbitrary length from an input word label, a discriminator $\mathcal{D}$, and a recognizer $\mathcal{R}$ that promotes style and data fidelity. While $\mathcal{D}$ promotes realistic-looking handwriting styles, $\mathcal{R}$ encourages the result to be readable. ScrabbleGAN minimizes a joint loss term $\mathcal{L} = \mathcal{L}_D + \lambda \mathcal{L}_R$ where $\mathcal{L}_D$ and $\mathcal{L}_R$ are the loss terms of $\mathcal{D}$ and $\mathcal{R}$, respectively, and the balance factor is $\lambda$. The generator $\mathcal{G}$ is designed such that each character is generated individually, using the property of the convolutions of overlapping receptive fields to account for the influence of nearby letters. Four character filters ($k_m$, $k_e$, $k_e$ and $k_t$) are concatenated, multiplied

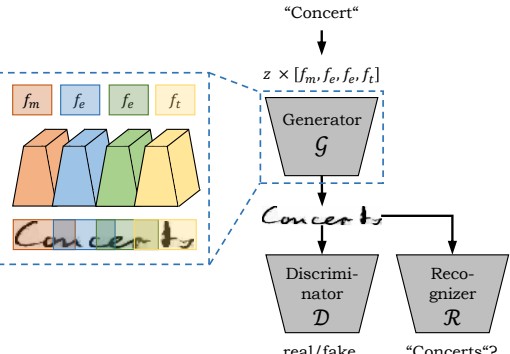

Figure 5: ScrabbleGAN concept by Fogel et al. (2020) of generating the word "Concerts".

by a noise vector $z$, and fed into a class-conditioned generator (see Figure 5). This allows for adjacent characters to interact and creates a smooth transition, e.g., enabling cursive text. The style of the image is controlled by a noise vector $z$ given as the input to the network (being consistent for all characters of a word). The recognizer $\mathcal{R}$ discriminates between real and gibberish text by comparing the output of $\mathcal{R}$ to the one that was given as input to $\mathcal{G}$. $\mathcal{R}$ is trained only on real and labeled samples. $\mathcal{R}$ is inspired by CRNN (Shi et al., 2017) and uses the CTC (Graves et al., 2009) loss. The architecture of the discriminator $\mathcal{D}$ is inspired by BigGAN (Brock et al., 2019) consisting of four residual blocks and a linear layer with one output. $\mathcal{D}$ is fully convolutional, predicts the average of the patches, and is trained with a hinge loss (Lim & Ye, 2017). We train ScrabbleGAN with the IAM-OffDB (Liwicki & Bunke, 2005) dataset and generate three different datasets. Exemplary images are shown in Figure 6. First, we generate 2 million images randomly selected from a large lexicon (*OffHW-German*), and pre-train the offline HWR architectures. Second, we generate 100,000 images based on the same word labels for each of the OnHW-words500 and OnHW-wordsRandom datasets (*OffHW-words500*, *OffHW-wordsRandom*]) and fine-tune the offline HWR architectures.

**Methods for Offline HWR.** OrigamiNet (Yousef & Bishop, 2020) is a state-of-the-art multi-line recognition method using only unsegmented image and text pairs. Similar to OrigamiNet, our offline method is based on different encoder architectures with one or two additional 1D convolutional layers (each with filter size 200, `softmax` activation (Zeng et al., 2017)) with 20% dropout for the latent representation, and a cross-modal representation decoder with BiLSTMs. For the encoder, we make use of Inception modules from GoogLeNet (Szegedy et al., 2015) and the ResNet34 (He

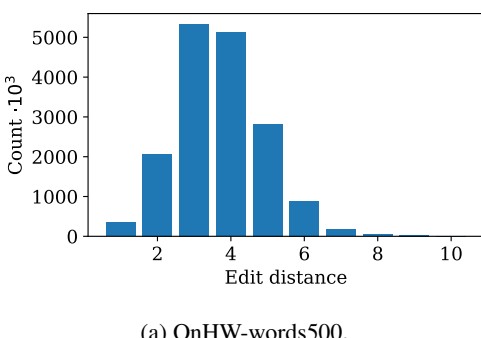

(a) OnHW-words500.

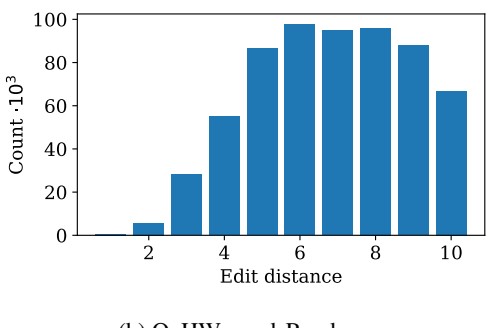

(b) OnHW-wordsRandom.

Figure 7: Number image-time-series pairs dependent on substitutions.

et al., 2016) architectures, and we re-implement the newly proposed gated, fully-convolutional method termed the gated text recognizer (Yousef et al., 2020). See Appendix A.5 for detailed information on the architectures. We train the networks on the generated OffHW-German dataset for 10 epochs and fine-tune on the OffHW-[500, wordsRandom] datasets for 15 epochs. For comparison with state-of-the-art techniques, we train OrigamiNet and compare with IAM-OffDB. For OrigamiNet, we apply interline spacing reduction via seam carving (Avidan & Shamir, 2007), resizing the images to 50% height, and random projective (rotating and resizing lines) and random elastic transform (Wigington et al., 2017). We augment the OffHW-German dataset with random width resizing and apply no augmentation for the OffHW-[words500, wordsRandom] datasets for fine-tuning.

**Offline/Online Cross-Modal Representation Learning.** Our architecture for online HWR is based on (Ott et al., 2022c). The encoder extracts features of the inertial data and consists of three convolutional layers (each with filter size 400, `ReLU` activation) and one convolutional layer (filter size 200, `ReLU` activation), a max pooling, batch normalization and a 20% dropout layer. As for the offline architecture, the network then learns a latent representation with one or two convolutional layers (each with filter size 200, `softmax` activation) with 20% dropout and the same cross-modal representation decoder. The output of the convolutional layers of the latent representation are minimized with the $\mathcal{L}_{\text{shared,c}}$ loss. The layers of the common representation are fine-tuned based on the pre-trained weights of the offline technique. Here, two BiLSTM layers with 60 units each and `ReLU` activation extract the temporal context of the feature embedding. As for the baseline classifier, we train for 1,000 epochs. For evaluation, the main time-series network is independent of the image auxiliary network by using only the weights of the main network.

**Triplet Selection.** To ensure (fast) convergence, it is crucial to select triplets that violate the constraint from Equation 1. Typically, it is infeasible to compute the loss for all triplet pairs, or this leads to poor training performance (as poorly chosen pairs dominate hard ones). This requires an elaborate triplet selection (Do et al., 2019). We use the Edit distance to define the identity and select triplets. The Edit distance is the minimum number of substitutions $S$, insertions $I$, and deletions $D$ required to change the sequences $\mathbf{d} = (d_1, \ldots, d_r)$ into $\mathbf{g} = (g_1, \ldots, g_z)$ with length $r$ and $z$, respectively. We define two sequences with an Edit distance of 0 as the positive pair, and with an Edit distance larger than 0 as the negative pair. Based on preliminary experiments, we use only substitutions for triplet selection that lead to a higher accuracy compared to additional insertions and deletions (whereas these would also change the length difference of image and time-series pairs). We constrain $p - m/2$ (the difference in pixels $p$ of the images and half the number of timesteps of the time-series) to be maximally $\pm 20$. The goal is to achieve a small distance for positive pairs and a large distance for negative pairs that increases with a larger Edit distance (between 1 and 10). Furthermore, despite a limited number of word labels, there still exist a large number of image-time-series pairs per word label for every possible Edit distance (see Figure 7). For each batch, we search in a dictionary of negative sample pairs for samples with $Edit\_distance = 1 + \lfloor \frac{\max_e - e - 1}{100} \rfloor$ as the lower bound for the current epoch $e$ and maximal epochs $\max_e$. For every label, we randomly pick one image. We let the margin $\alpha$ in the triplet loss vary for each batch such that $\alpha = \beta \cdot \overline{Edit\_distance}$ depends on the mean Edit distance of the batch and is in the range $[1, 11]$ with $\beta = 10^{-3}$ for MSE, $\beta = 0.1$ for CS and PC, and $\beta = 1$ for KL. The batch size is 100 and $\max_e = 1,000$.

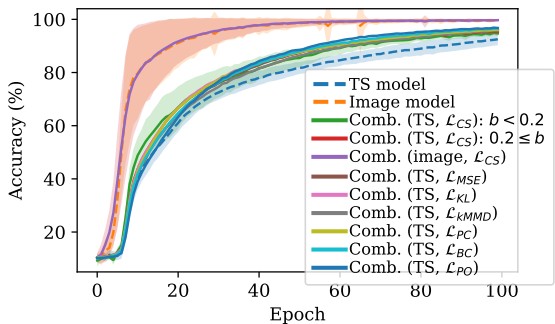

| Method | Accuracy (%) |
|---|---|
| TS model | 92.50 |
| Combined (TS, $\mathcal{L}_{CS}$) | 95.36 |
| Combined (image, $\mathcal{L}_{CS}$) | 99.70 |
| Combined (TS, $\mathcal{L}_{MSE}$) | 96.25 |
| Combined (TS, $\mathcal{L}_{KL}$) | 96.22 |
| Combined (TS, $\mathcal{L}_{kMMD}$) | 95.83 |
| Combined (TS, $\mathcal{L}_{PC}$) | 96.03 |
| Combined (TS, $\mathcal{L}_{BC}$) | 96.62 |
| Combined (TS, $\mathcal{L}_{PO}$) | **96.76** |

Figure 8: Accuracy of single- and cross-modal representation learning over all epochs.

Table 3: Comparison of single- and cross-modal representation learning.

## 5 Experimental Results

**Hardware and Training Setup.** For all experiments, we use Nvidia Tesla V100-SXM2 GPUs with 32 GB VRAM equipped with Core Xeon CPUs and 192 GB RAM. We use the vanilla Adam optimizer with a learning rate of $10^{-4}$.

### 5.1 Evaluation of Synthetic Data

We train the time-series (TS) model 18 times with noise $b = 0.3$ and the combined model with the triplet loss for all 40 noise combinations $\left( b \in \{0, \dots, 1.95\} \right)$ with different deep metric learning functions. Figure 8 shows the validation accuracy averaged over all trainings as well as the combined cases separately for noise $b < 0.2$ and noise $0.2 \leq b < 2.0$ (for the $\mathcal{L}_{CS}$ loss). Table 3 summarizes the final classification results of all cases. The accuracy of the models that use only images and in combination with time-series during inference reach an accuracy of 99.7% (which can be seen as an unreachable upper bound for the TS-only models). The triplet loss improves the final TS baseline accuracy from 92.5% to 95.36% (averaged over all combinations), while combining TS and image data leads to a faster convergence. Conceptually similar to (Long et al., 2015), we use the $\mathcal{L}_{kMMD}$ loss, which yields 95.83% accuracy. The $\mathcal{L}_{PC}$ (96.03%), $\mathcal{L}_{KL}$ (96.22%), $\mathcal{L}_{MSE}$ (96.25%), $\mathcal{L}_{BC}$ (96.62%), and $\mathcal{L}_{PO}$ (96.76%) loss functions can further improve the accuracy. We conclude that the triplet loss can be successfully used for cross-modal learning by utilizing negative identities.

### 5.2 Evaluation of Handwriting Recognition

**Evaluation Metrics.** A metric for sequence evaluation is the character error rate (CER), defined as CER $= \frac{S_c + I_c + D_c}{N_c}$, i.e., the Edit distance (the sum of character substitutions $S_c$, insertions $I_c$ and deletions $D_c$) divided by the total number of characters in the set $N_c$. Similarly, the word error rate (WER) is defined as WER $= \frac{S_w + I_w + D_w}{N_w}$, which is computed with the sum of word operations $S_w$, $I_w$ and $D_w$, divided by the number of words in the set $N_w$.

**Evaluation of Offline HWR Methods.** Table 4 shows offline HWR results on our generated OffHW-German dataset and on the IAM-OffDB (Liwicki & Bunke, 2005) dataset. ScrabbleGAN (Fogel et al., 2020) yields a WER of 23.61% on the IAM-OffDB dataset, while OrigamiNet (Yousef & Bishop, 2020) achieves a CER of 4.70% with 12 gated text recognizer modules. While OrigamiNet is trained for the multi-line classification, which is an easier task (as the image of the paragraph does not have to be segmented into lines), we trained OrigamiNet on single-lines with zero padding, which is closer to the OffHW-German dataset. While the images for the multi-line task are of approximately similar lengths, the image lengths of the single-line task varies strongly, and hence, zero padding has a high influence on the model performance, resulting in a CER of 15.67%. While Yousef & Bishop (2020) did not propose WER results, OrigamiNet yields only a WER of 90.40%. This problem does not appear for the OffHW-German dataset, as the dataset contains only single words with similar lengths. With our own implementation of four gated text recognizer modules and one convolutional layer for the common representation, our model achieves similar results. As the training takes more than one day for one epoch on the large OffHW-German dataset, we train OrigamiNet with four gated text recognizer modules, and achieve 0.11% CER on the generated dataset and 15.67% on the IAM-OffDB

Table 4: Evaluation results (WER and CER in %) for the generated dataset with ScrabbleGAN (Fogel et al., 2020) OffHW-German and the IAM-OffDB (Liwicki & Bunke, 2005) dataset.

| | | OffHW-German | | IAM-OffDB | |
|---|---|---|---|---|---|
| | **Method** | **WER** | **CER** | **WER** | **CER** |
| **Related Work Work** | ScrabbleGAN (Fogel et al., 2020) | - | - | 23.61 | - |
| | OrigamiNet ($12 \times$ gated text recognizer) (Yousef & Bishop, 2020) | - | - | - | 4.70 |
| **Our Implementation** | OrigamiNet (ours, $4 \times$ gated text recognizer) | 1.50 | 0.11 | 90.40 | 15.67 |
| | Inception | 12.54 | 1.17 | - | - |
| | ResNet | 13.05 | 1.24 | - | - |
| | Gated text recognizer (2 blocks), 1 conv. layer | 4.34 | 0.39 | - | - |
| | Gated text recognizer (2 blocks), 2 conv. layer | 5.02 | 0.44 | - | - |
| | Gated text recognizer (4 blocks), 1 conv. layer | 3.35 | 0.34 | 89.37 | 15.60 |
| | Gated text recognizer (4 blocks), 2 conv. layer | 2.52 | 0.24 | - | - |
| | Gated text recognizer (6 blocks) | 2.85 | 0.26 | - | - |
| | Gated text recognizer (8 blocks) | 4.22 | 0.38 | - | - |

Table 5: Evaluation results (WER and CER in %) for the generated OffHW-words500 and OffHW-wordsRandom datasets for one and two convolutional layers (c). We propose writer-dependent (WD) and writer-independent (WI) results.

| Method | OffHW-words500 | | | | OffHW-wordsRandom | | | |
|---|---|---|---|---|---|---|---|---|
| | WD | | WI | | WD | | WI | |
| ($4 \times$ gated text recognizer) | **WER** | **CER** | **WER** | **CER** | **WER** | **CER** | **WER** | **CER** |
| $c = 1$ | 2.94 | 0.76 | 0.95 | 0.23 | 1.98 | 0.35 | 2.05 | 0.37 |
| $c = 2$ | 2.51 | 0.69 | 0.85 | 0.22 | 1.82 | 0.34 | 1.95 | 0.38 |

dataset. All our models yield low error rates on the generated OffHW-German dataset. Our approach with gated text recognizer blocks outperforms (0.24% to 0.44% CER) the models with Inception (Szegedy et al., 2015) (1.17% CER) and ResNet (He et al., 2016) (1.24% CER). OrigamiNet achieves the lowest error rates of 1.50% WER and 0.11% CER. Four gated text recognizer blocks yield the best results at a significantly lower training time compared to six or eight blocks. We fine-tune the model with four gated text recognizer blocks for one and two convolutional layers and achieve notably low error rates between 0.22% to 0.76% CER, and between 0.85% to 2.95% WER on the OffHW-[words500, wordsRandom] datasets (see Table 5). While results for OffHW-wordsRandom are similar for writer-dependent (WD) and writer-independent (WI) tasks, WI results of the OffHW-words500 dataset are lower than WD results, as words with the same label appear in the training and test dataset. We use the weights of the fine-tuning as initial weights of the image model for the cross-modal representation learning.

**Evaluation of Representation Learning Feature Embeddings.** Table 6 shows the feature embeddings for image $f_2(\mathbf{X}_i)$ and time-series data $f_2(\mathbf{U}_i)$ of the *positive* sample Export and the two *negative* samples Expert ($Edit\_distance = 1$) and Import ($Edit\_distance = 2$) based on four deep metric learning loss functions. The pattern of characters are similar, as the words differ only in the fourth letter. In contrast, Import has a different feature embedding, as the replacement of E with I and x with m leads to a higher feature distance in the embedding hypersphere. Note that image and time-series data can vary in length for $Edit\_distance > 0$. Figure 9 shows the feature embeddings of the output of the convolutional layers ($c = 1$) processed with t-SNE (van der Maaten & Hinton, 2008). Figure 9a visualizes the multivariate time-series embeddings $f_1(\mathbf{U}_i)$ of the single modal network. The learned representation generalizes well, but misclassifications (e.g., of small and capital letters at the beginning of a word, which happen quite often) also introduce errors in the latent representation. Figure 9b visualizes the multivariate time-series and image embeddings $\left(f_1(\mathbf{U}_i) \text{ and } f_1(\mathbf{X}_i), \text{ respectively}\right)$ in a cross-modal setup. While the embedding of the single modal network is unstructured, the embeddings of the cross-modal network are structured (distance of

Table 6: Feature embeddings $f_c(\mathbf{X}_i)$ and $f_c(\mathbf{U}_i)$ of exemplary image $\mathbf{X}_i$ and multivariate time-series $\mathbf{U}_i$ data of the convolutional layer $c = \text{conv}_2$ for different deep metric learning functions for positive pairs ($Edit\_distance = 0$) and negative pairs ($Edit\_distance > 0$) trained with the triplet loss. The feature embeddings are similar in the red box (character x) or blue box (character p) for $f_2(\mathbf{X}_i)$, or the last pixels (character t) of $f_2(\mathbf{U}_i)$ for $\mathcal{L}_{\text{PC}}$ marked green.

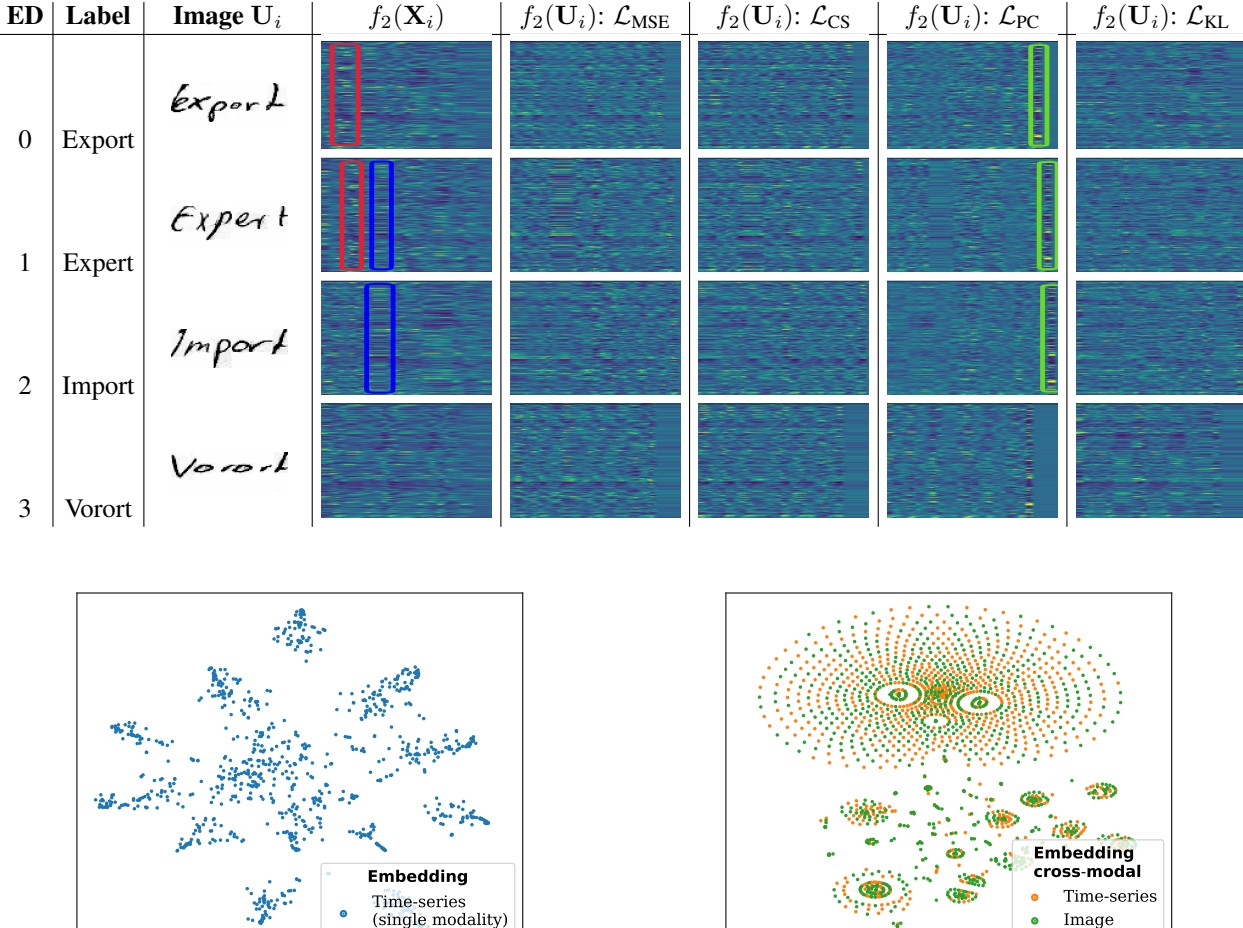

(a) Feature embedding of IMU samples for the single modalitiy network.

(b) Feature embeddings of IMU and image samples for the cross-modal network.

Figure 9: Comparison of the naive method (left) and our proposed approach (right), where our method shows a much better behaved embedding space compared to the naive approach by learning a joint representation. Plot of $400 \times 200$ feature embeddings of image and IMU modalities with t-SNE.

samples visualizes the Edit distance between words) with the embeddings of the time-series modality being close to the embeddings of the image modality, and hence, more distinctive clusters with better separation.

**Evaluation of Cross-Modal Representation Learning.**   Table 7 gives an overview of cross-modal representation learning (for $c = 1$). The first row shows baseline results by Ott et al. (2022c): 13.04% CER on OnHW-words500 (WD) and 6.75% CER on OnHW-wordsRandom (WD) with mutated vowels. Compared to various time-series classification techniques, their benchmark results showed superior performance of CNN+BiLSTMs on these OnHW recognition tasks. Only InceptionTime (Fawaz et al., 2019) (a large time-series encoder network with $depth = 11$ and $nf = 96$) – with BiLSTM layers – yields partly better results or is on par with the CNN+BiLSTM model for sequence-based classification, while the CNN+BiLSTM model outperforms state-of-the-art techniques on single character-based classification tasks. Due to the faster training of the CNN+BiLSTM, we chose this network for the cross-modal task.

Table 7: Evaluation results (WER and CER in %) averaged over five splits of the baseline time-series-only technique and our cross-modal learning technique for the inertial-based OnHW datasets (Ott et al., 2022c) with and without mutated vowels (MV) for one convolutional layer $c = 1$. Best results are **bold**, and second best results are underlined. Arrows indicate improvements ($\uparrow$) and degradation ($\downarrow$) of baseline results (w/o MV).

| | Method | OnHW-words500 | | | | OnHW-wordsRandom | | | |
| | | WD | | WI | | WD | | WI | |
| | | WER | CER | WER | CER | WER | CER | WER | CER |
|---|---|---|---|---|---|---|---|---|---|
| **Main Task** | InceptionTime, $\mathcal{L}_{\text{CTC}}$, w/ MV | 37.12 | 12.96 | 62.09 | 26.36 | 42.88 | 7.19 | 84.14 | 32.35 |
| | IT+BiLSTM, $\mathcal{L}_{\text{CTC}}$, w/ MV | 43.22 | 13.07 | 61.62 | 26.08 | 39.14 | 6.39 | 85.42 | 33.31 |
| | CNN+BiLSTM, $\mathcal{L}_{\text{CTC}}$, w/ MV | 42.81 | 13.04 | 60.47 | 28.30 | 37.13 | 6.75 | 83.28 | 35.90 |
| | CNN+BiLSTM, $\mathcal{L}_{\text{CTC}}$, w/o MV | 42.77 | 13.44 | 59.82 | 28.54 | 38.02 | 7.81 | 83.54 | 36.51 |
| **Baseline** | $\mathcal{L}_{\text{MSE}}$ | 40.76 $\uparrow$ | 12.71 $\uparrow$ | **55.54** $\uparrow$ | 25.97 $\uparrow$ | 37.31 $\uparrow$ | 7.01 $\uparrow$ | 82.25 $\uparrow$ | 33.85 $\uparrow$ |
| | $\mathcal{L}_{\text{CS}}$ | 38.62 $\uparrow$ | 11.55 $\uparrow$ | 56.37 $\uparrow$ | **25.90** $\uparrow$ | 38.85 $\downarrow$ | 7.35 $\uparrow$ | 82.48 $\uparrow$ | 35.67 $\uparrow$ |
| | $\mathcal{L}_{\text{PC}}$ | 39.09 $\uparrow$ | 11.69 $\uparrow$ | 57.90 $\uparrow$ | 27.23 $\uparrow$ | 38.46 $\downarrow$ | 7.15 $\uparrow$ | 82.71 $\uparrow$ | 35.13 $\uparrow$ |
| | $\mathcal{L}_{\text{KL}}$ | 38.36 $\uparrow$ | 11.28 $\uparrow$ | 60.23 $\downarrow$ | 27.99 $\uparrow$ | 38.76 $\downarrow$ | 7.49 $\uparrow$ | **81.07** $\uparrow$ | 33.96 $\uparrow$ |
| **Contrastive Loss** | $\mathcal{L}_{\text{contr},1}(\mathcal{L}_{\text{MSE}})$ | 38.34 $\uparrow$ | 11.57 $\uparrow$ | 56.81 $\uparrow$ | 25.98 $\uparrow$ | 38.25 $\downarrow$ | 7.31 $\uparrow$ | 82.09 $\uparrow$ | 34.03 $\uparrow$ |
| | $\mathcal{L}_{\text{contr},1}(\mathcal{L}_{\text{CS}})$ | 39.68 $\uparrow$ | 11.73 $\uparrow$ | 58.03 $\uparrow$ | 27.13 $\uparrow$ | **35.96** $\uparrow$ | **6.67** $\uparrow$ | 81.22 $\uparrow$ | 33.11 $\uparrow$ |
| | $\mathcal{L}_{\text{contr},1}(\mathcal{L}_{\text{PC}})$ | 37.82 $\uparrow$ | 11.34 $\uparrow$ | 57.45 $\uparrow$ | 26.18 $\uparrow$ | 39.22 $\downarrow$ | 7.39 $\uparrow$ | 82.45 $\uparrow$ | 34.21 $\uparrow$ |
| | $\mathcal{L}_{\text{contr},1}(\mathcal{L}_{\text{KL}})$ | **36.70** $\uparrow$ | **10.84** $\uparrow$ | 61.72 $\downarrow$ | 29.16 $\downarrow$ | 38.92 $\downarrow$ | 7.51 $\uparrow$ | 83.54 | 35.52 $\uparrow$ |
| **Triplet Loss** | $\mathcal{L}_{\text{trpl},1}(\mathcal{L}_{\text{MSE}})$ | 42.95 $\downarrow$ | 14.13 $\downarrow$ | 56.48 $\uparrow$ | 26.66 $\uparrow$ | 37.66 $\uparrow$ | 7.04 $\uparrow$ | 81.64 $\uparrow$ | 34.39 $\uparrow$ |
| | $\mathcal{L}_{\text{trpl},1}(\mathcal{L}_{\text{CS}})$ | 38.01 $\uparrow$ | 11.29 $\uparrow$ | 58.50 $\uparrow$ | 27.10 $\uparrow$ | 37.12 $\uparrow$ | 6.98 $\uparrow$ | 82.71 $\uparrow$ | **33.09** $\uparrow$ |
| | $\mathcal{L}_{\text{trpl},1}(\mathcal{L}_{\text{PC}})$ | 40.43 $\uparrow$ | 12.41 $\uparrow$ | 58.20 $\uparrow$ | 27.48 $\uparrow$ | 37.40 $\uparrow$ | 7.01 $\uparrow$ | 81.90 $\uparrow$ | 33.89 $\uparrow$ |
| | $\mathcal{L}_{\text{trpl},1}(\mathcal{L}_{\text{KL}})$ | 37.55 $\uparrow$ | 11.21 $\uparrow$ | 63.52 $\downarrow$ | 30.52 $\downarrow$ | 38.39 $\downarrow$ | 7.36 $\uparrow$ | 83.18 $\uparrow$ | 35.21 $\uparrow$ |

In general, the word error rate (WER) can vary for a similar character error rate (CER). The reason is that a change of one character of a correctly classified word leads to a large change in the WER, while the change of the CER is marginal. We define the results trained without mutated vowels as baseline results, as ScrabbleGAN is pretrained on IAM-OffDB, which does not contain mutated vowels, and hence, such words cannot be generated. Nevertheless, the main model can be trained and is applicable to samples with mutated vowels. For a fair comparison, we compare our results to the results of the models trained without mutated vowels. Here, the error rates are slightly higher for both datasets. As expected, cross-modal learning improves the baseline results up to 11.28% CER on the OnHW-words500 WD dataset and up to 7.01% CER on the OnHW-wordsRandom WD dataset. The contrastive loss shows the best results on the OnHW-words500 (WD) dataset with the Kullback-Leibler metric and on the OnHW-wordsRandom dataset (WD) with the cosine similarity metric. With the triplet loss, $\mathcal{L}_{\text{CS}}$ outperforms other metrics on the OnHW-wordsRandom dataset but is inconsistent on the OnHW-words500 dataset. The importance of the triplet loss is more significant for one convolutional layer ($c = 1$) than for two convolutional layers ($c = 2$) (see Appendix A.6). Furthermore, training with kMMD (implemented as in (Long et al., 2015)) does not yield reasonable results. We assume that this metric cannot make use of the important time component in the HWR application. We proposed our approach as learning with privileged information by exploiting a visual modality as an auxiliary task and improve the main task based on an inertial modality. The cross-modal learning would also work for the visual modality as the main task and a generated dataset for the inertial modality as an auxiliary task. However, the error rates are already low for the image-based classification task, as methods for offline HWR are very advanced and the image dataset is very large. Hence, we assume that fine-tuning the image encoder with inertial data would result in a minor improvement. Prior work Ott et al. (2022c) evaluated data augmentation techniques for multivariate time-series data (i.e., time warping, scaling, jittering, magnitude warping, and shifting). This approach was rather limited with only 2-3% points of improvement compared with augmentation with the auxiliary image-based task.

**Transfer Learning on Left-Handed Writers.** To adapt the model to left-handed writers (who are typically underrepresented and hence marginalized in the real-world), we make use of the left-handed datasets OnHW-words500-L and OnHW-wordsRandom-L proposed by Ott et al. (2022c). These datasets contain recordings of two writers who provided 1,000 and 996 samples. As a baseline, we pre-train the time-series-only model on the right-handed datasets and post-train the left-handed datasets for 500 epochs (see the second and third rows of Table 8). As these datsets are

Table 8: Evaluation results (WER and CER in %) averaged over five splits of the baseline time-series-only technique and our cross-modal techniques for the inertial-based left-handed writers OnHW datasets (Ott et al., 2022c) with and without mutated vowels (MV) for one ($c = 1$) and two ($c = 2$) convolutional layers $c = 1$. Best results are **bold**, and second best results are underlined. Arrows indicate improvements ($\uparrow$) and degradation ($\downarrow$) of baseline results (w/o MV).

| | | OnHW-words500-L | | | | OnHW-wordsRandom-L | | | |
| | | WD | | WI | | WD | | WI | |
| | **Method** | **WER** | **CER** | **WER** | **CER** | **WER** | **CER** | **WER** | **CER** |
|---|---|---|---|---|---|---|---|---|---|
| **Main Task** | InceptionTime, $\mathcal{L}_{\text{CTC}}$, w/ MV | 49.70 | 14.02 | 100.0 | 96.06 | 48.10 | 8.63 | 100.0 | 95.93 |
| | CNN+BiLSTM, $\mathcal{L}_{\text{CTC}}$, w/ MV | 14.20 | 3.30 | 94.40 | 71.41 | 30.20 | 4.86 | 100.0 | 83.51 |
| | CNN+BiLSTM, $\mathcal{L}_{\text{CTC}}$, w/o MV | 12.94 | 3.33 | 89.07 | 62.07 | 30.89 | 5.26 | 100.0 | 81.15 |
| **Baseline** | $\mathcal{L}_{\text{MSE}}$ | 11.62 $\uparrow$ | 2.77 $\uparrow$ | 90.65 $\downarrow$ | 67.90 $\downarrow$ | 30.53 $\uparrow$ | 4.93 $\uparrow$ | 100.0 | 81.99 $\downarrow$ |
| | $\mathcal{L}_{\text{CS}}$ | 14.92 $\downarrow$ | 3.53 $\downarrow$ | 94.14 $\downarrow$ | 65.10 $\downarrow$ | 29.06 $\uparrow$ | 4.87 $\uparrow$ | 100.0 | 83.94 $\downarrow$ |
| | $\mathcal{L}_{\text{PC}}$ | 12.29 $\uparrow$ | 3.04 $\uparrow$ | 91.33 $\downarrow$ | 60.89 $\uparrow$ | 27.32 $\uparrow$ | **4.47** $\uparrow$ | 100.0 | 85.09 $\downarrow$ |
| | $\mathcal{L}_{\text{KL}}$ | **11.37** $\uparrow$ | **2.57** $\uparrow$ | 93.02 $\downarrow$ | 66.64 $\downarrow$ | 29.61 $\uparrow$ | 4.91 $\uparrow$ | 100.0 | 81.28 $\downarrow$ |
| **Triplet Loss** | $\mathcal{L}_{\text{trpl},1}(\mathcal{L}_{\text{MSE}})$ | 12.48 $\uparrow$ | 3.11 $\uparrow$ | 90.09 $\downarrow$ | 62.87 $\downarrow$ | 32.62 $\downarrow$ | 5.43 $\downarrow$ | 100.0 | **80.41** $\uparrow$ |
| | $\mathcal{L}_{\text{trpl},1}(\mathcal{L}_{\text{CS}})$ | 13.65 $\downarrow$ | 3.28 $\uparrow$ | 90.76 $\downarrow$ | 62.40 $\downarrow$ | 34.21 $\downarrow$ | 5.53 $\downarrow$ | 100.0 | 82.14 $\downarrow$ |
| | $\mathcal{L}_{\text{trpl},1}(\mathcal{L}_{\text{PC}})$ | 13.71 $\downarrow$ | 3.23 $\uparrow$ | 91.55 $\downarrow$ | 65.95 $\downarrow$ | 31.59 $\downarrow$ | 5.32 $\downarrow$ | 100.0 | 81.77 $\downarrow$ |
| | $\mathcal{L}_{\text{trpl},1}(\mathcal{L}_{\text{KL}})$ | 13.65 $\downarrow$ | 3.45 $\downarrow$ | 94.93 $\downarrow$ | 72.01 $\downarrow$ | 31.87 $\downarrow$ | 5.42 $\downarrow$ | 100.0 | 82.02 $\downarrow$ |
| | $\mathcal{L}_{\text{trpl},2}(\mathcal{L}_{\text{MSE}})$ | 11.97 $\uparrow$ | 2.83 $\uparrow$ | **84.34** $\uparrow$ | **57.84** $\uparrow$ | **27.19** $\uparrow$ | 4.79 $\uparrow$ | **99.87** $\uparrow$ | 82.60 $\downarrow$ |
| | $\mathcal{L}_{\text{trpl},2}(\mathcal{L}_{\text{CS}})$ | 11.65 $\uparrow$ | 2.63 $\uparrow$ | 94.70 $\downarrow$ | 67.69 $\downarrow$ | 28.39 $\uparrow$ | 4.62 $\uparrow$ | 100.0 | 83.44 $\downarrow$ |
| | $\mathcal{L}_{\text{trpl},2}(\mathcal{L}_{\text{PC}})$ | 13.02 $\downarrow$ | 2.94 $\uparrow$ | 89.86 $\downarrow$ | 60.26 $\uparrow$ | 30.22 $\uparrow$ | 4.81 $\uparrow$ | 100.0 | 84.29 $\downarrow$ |
| | $\mathcal{L}_{\text{trpl},2}(\mathcal{L}_{\text{KL}})$ | 13.55 $\downarrow$ | 3.22 $\uparrow$ | 97.86 $\downarrow$ | 76.54 $\downarrow$ | 28.14 $\uparrow$ | 4.71 $\uparrow$ | 100.0 | 80.81 $\uparrow$ |

rather small, the models can overfit on these specific writers and achieve a very low CER of 3.33% on the OnHW-words500-L datasets and 5.26% CER on the OnHW-wordsRandom-L dataset without mutated vowels for the writer-dependent tasks. However, the models cannot generalize on the writer-independent tasks, as evidenced by 62.07% CER on the OnHW-words500-L dataset and 81.15% CER on the OnHW-wordsRandom-L dataset. Hence, we focus on the WD tasks. For comparison, we use the state-of-the-art time-series classification technique InceptionTime (Fawaz et al., 2019) with $depth = 11$ and $nf = 96$ (without pre-training). As shown, our CNN+BiLSTM outperforms InceptionTime by a considerable margin. We use the weights of the pre-training with the offline handwriting datasets and again post-train on the left-handed datasets with $c = 1$ and $c = 2$. Using the weights of the cross-modal learning without the triplet loss can decrease the error rates up to 2.57% CER with $\mathcal{L}_{\text{KL}}$ and 4.47% CER with $\mathcal{L}_{\text{PC}}$. Using the triplet loss $\mathcal{L}_{\text{trpl},2}(\mathcal{L}_{\text{MSE}})$ can further significantly decrease the WI OnHW-words500-L error rates. In conclusion, due to the use of the weights of the cross-modal setup, the model can adapt faster to new writers and generalize better to unseen words due to the triplet loss.

# 6 Conclusion

We evaluated metric learning-based triplet loss functions for cross-modal representation learning between image and time-series modalities with class label-specific triplet selection. On synthetic data as well as on different HWR datasets, our method yields notable accuracy improvements for the main time-series classification task and can be decoupled from the auxiliary image classification task at inference time. Our cross-modal triplet selection further yields a faster training convergence with better generalization on the main task.

**Broader Impact Statement**

While research for offline handwriting recognition (HWR) is well-established, research for online HWR from sensor-enhanced pens only emerged in 2019. Hence, the methodological research for online HWR currently does not meet the requirements for real-world applications. Handwriting is still important in different fields, in particular graphomotoricity as a fine motor skill. The visual feedback provided by the pen helps young students to learn a new language. A well-known bottleneck for many machine learning algorithms is their requirement for large amounts of datasets, while

data recording of handwriting data is time-consuming. This paper extends the online HWR dataset with generated images from offline handwriting and closes the gap between offline and online HWR by using offline HWR as an auxiliary task by learning with privileged information. One downside of training the offline architecture (consisting of gated text recognizer blocks) is its long training time. However, as this model is not required at inference time, processing the time-series is still fast. The cross-modal representation between both modalities (image and time-series) is achieved by using the triplet loss and a sample selection depending on the Edit distance. This approach is important in many applications of sequence-based classification, i.e., the triplet loss evolved recently for language processing applications such as visual semantic clustering, while pairwise learning is typically applied in fields such as image recognition. Ethical statement about collection consent and personal information: For data recording, the consent of all participants was collected. The datasets only contain the raw data from the sensor-enhanced pen and – for statistics – the age, gender, and handedness of the participants. The datasets are fully pseudonymized by assigning an ID to every participant. The datasets do not contain any personal identifying information. The approach proposed in this paper – in particular, when used for the application of online handwriting recognition from sensor-enhanced pens – does not (1) facilitate injury to living beings, (2) raise safety or security concerns (due to the anonymity of the data), (3) raise human rights concerns, (4) have a detrimental effect on people's livelihood, (5) develop harmful forms of surveillance (as the data is pseudonymized), (6) damage the environment, and (7) deceive people in ways that cause harm.

### Limitations

The limitation of the method is the requirement of multiple image-based datasets in the same language. As the OnHW-words and OnHW-wordsRandom datasets are written in German and contain word labels with mutated vowels, a similar image-based German dataset is required, which does not currently exist. The available dataset most similar to the OnHW dataset is the IAM-OffDB dataset, which does not contain mutated vowels. Hence, the OCR method cannot be pre-trained on words with mutated vowels. In conclusion, the method is not limited by ScrabbleGAN, but by the image-based dataset required for pre-training. The gated text recognizer could also be directly pre-trained on the IAM-OffDB dataset, but we assume less generalized results than for our generated dataset.

### Statement on Ethical Concerns

Machine learning models face various challenges when classifying text with this sensor-enhanced pen. These challenges can appear if there is a domain shift between training and test datasets, e.g., specific writers have a unique writing style and accelerations, or they hold the pen differently. Also, some writers might have a unique writing environment (different writing surfaces such as a unique table or paper which leads to different magnetic fields). Another difficulty can appear through an under-represented group such as left-handed writers or a disabled person for which the model is not trained on. A well-generalized model trained on all possible pen movements is very challenging and requires a lot of training data. One solution is to record data for a unique writer and adapt the model, or augment the data for a better representation, e.g., as proposed with our method on left-handed writers. Hence, unique writers are not excluded and the task for classifying writing from under-represented groups is addressed in our paper, but domain shifts still remain a challenging problem.

### Comparison to Writing on Touch Screen Surfaces

Methods for writing on surfaces such as the iPad OS system and others require a tablet with a touch screen surface and stylus pens with integrated magnetometers or pressure sensitivity. These methods can easily reconstruct the trajectory of the pen tip through the magnetometer on the surface, and hence, can classify the written text. This is more challenging when using sensor-enhanced pens, as the classification task is performed directly on the sensor data. One drawback of methods used in the iPad OS is the requirement for writing on specific surfaces, which in turn can influence the writing style. Also, certain applications require writing on normal paper, or the availability of a touch screen surface is not always given, e.g., when writing a short list, but notes need to be digitized afterwards.

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

# A  Appendices

While Section A.1 gives an overview of methods for offline handwriting recognition, Section A.2 summarizes cross-modal retrieval methods, the corresponding modalities, pairwise learning, and deep metric learning. We present the multi-task learning technique in Section A.3, and show more details on learning with the triplet loss on synthetically generated signal and image data in Section A.4. We propose more details of our architectures in Section A.5. Section A.6 presents results of representation learning for online HWR.

## A.1  Offline Handwriting Recognition

In the following, we give a detailed overview of offline HWR methods to select a suitable lexicon and language model-free method. To our knowledge, there is no recent paper summarizing published work for offline HWR. For an overview of offline and online HWR datasets, see (Plamondon & Srihari, 2000; Hussain et al., 2015). Table 9 presents related work. Methods for offline HWR range from hidden Markov models (HMMs) to deep learning techniques that became predominant in 2014, such as convolutional neural networks (CNNs), temporal convolutional networks (TCNs), and recurrent neural networks (RNNs). RNN techniques are well explored, including long short-term memories (LSTMs), bidirectional LSTMs (BiLSTMs), and multidimensional RNNs (MDRNN, MDLSTM). Recent methods are generative adversarial networks (GANs) and Transformers. In Table 9, we refer to the use of a language model as LM with $k$ and identify the data level on which the method works – i.e., paragraph or full-text level (P), line level (L), and word level (W). We present evaluation results for the IAM-OffDB (Liwicki & Bunke, 2005) and RIMES (Grosicki & El-Abed, 2011) datasets. We show the character error rate (CER) – the percentage of characters that were incorrectly predicted (the lower, the better) – and the word error rate (WER) – a common performance metric on word level instead of the phoneme level (the lower, the better).

**HMMs.**  In the past, various methods based on HMMs have been proposed (Bertolami & Bunke, 2018; Dreuw et al., 2011; Li et al., 2014; Pastor-Pellicer et al., 2015). Recently, España-Boquera et al. (2011) proposed HMM+ANN, an HMM modeled with Markov chains in combination with a multilayer perceptron (MLP) to estimate the emission probabilities. Kozielski et al. (2013) presented Tandem GHMM that uses moment-based image normalization, writer adaptation, and discriminative feature extraction with a 3-gram open-vocabulary of size 50k with an LSTM for recognition. Doetsch et al. (2014) proposed an LSTM unit that controls the shape of the squashing function in gating units decoded in a hybrid HMM. This approach yields the best results based on HMMs.

**RNNs: MDLSTMs.**  The 2DLSTM approach by Graves & Schmidhuber (2008) combines multidimensional LSTMs (MDLSTMs) with the CTC loss. The MDLSTM-RNN approach (Bluche, 2016) works at paragraph level by replacing the collapse layer with a recurrent version. A neural network performs implicit line segmentation by computing attention weights on the image representation. Voigtlaender et al. (2016) proposed an efficient GPU-based implementation of MDLSTMs by processing the input in a diagonal-wise fashion. SepMDLSTM (Chen et al., 2017) is a multi-task learning method for script identification and HWR based on two classification modules by minimizing the CTC and negative log-likelihood losses. While the MDLSTM by Bluche et al. (2017) contains covert and overt attention without prior segmentation, the Castro et al. (2018) integrated MDLSTMs within a hybrid HMM. However, these architectures come with expensive computational cost. Furthermore, they extract features visually similar to those of convolutional layers (Puigcerver, 2017). End2End (Krishnan et al., 2018) jointly learns text and image embeddings based on LSTMs.

**RNNs: LSTMs and BiLSTMs.**  RNNs for HWR marked an important milestone in achieving impressive recognition accuracies. Sequential architectures are perfect to fit text lines, due to the probability distributions over sequences of characters, and due to the inherent temporal aspect of text (Kang et al., 2022). Graves et al. (2009) introduced the BiLSTM layer in combination with the CTC loss. Pham et al. (2014) showed that the performance of LSTMs can be greatly improved using dropout. Voigtlaender et al. (2015) investigated sequence-discriminative training of LSTMs using the maximum mutual information (MMI) criterion. While Bluche (2015) utilized an RNN with an HMM and a language model, Menasri et al. (2012) combined an RNN with a sliding window Gaussian HMM. GCRNN (Bluche & Messina, 2017) combines a convolutional encoder (aiming for generic and multilingual features) and a BiLSTM decoder predicting character sequences. Additionally, Puigcerver (2017) proposed a CNN+BiLSTM architecture (CNN-1DLSTM-CTC) that uses the CTC loss. The start, follow, read (SFR) (Wigington et al., 2018) model jointly learns text detection and segmentation. Dutta et al. (2018) used synthetic data for pre-training and image normalization for

Table 9: Evaluation results (WER and CER in %) of different methods on the IAM-OffDB (Liwicki & Bunke, 2005) and RIMES (Grosicki & El-Abed, 2011) datasets. The table is sorted by year.

| | Method | Information | LM size $k$ | P | L | W | IAM-OffDB WER | IAM-OffDB CER | RIMES WER | RIMES CER |
|---|---|---|---|---|---|---|---|---|---|---|
| **HMM** | HMM+ANN (España-Boquera et al., 2011) | Markov chain with MLP | w/ (5) | | | | 15.50 | 6.90 | - | - |
| | Tandem GHMM (Kozielski et al., 2013) | GHMM and LSTM, writer adaptation | w/ (50) | | | × | 13.30 | 5.10 | 13.70 | 4.60 |
| | LSTM-HMM (Doetsch et al., 2014) | Combination of LSTM with HMM | w/ (50) | | × | | 12.20 | 4.70 | 12.90 | 4.30 |
| **Multi-dim. LSTM** | 2DLSTM (Graves & Schmidhuber, 2008) | Combined MDLSTM with CTC | w/o | | | | 27.50 | 8.30 | 17.70 | 4.00 |
| | MDLSTM-RNN (Bluche, 2016) | 150 dpi | w/o | × | | | 29.50 | 10.10 | 13.60 | 3.20 |
| | | 150 dpi | w/ (50) | × | | | 16.60 | 6.50 | - | - |
| | | 300 dpi | w/o | × | | | 24.60 | 7.90 | 12.60 | 2.90 |
| | | 300 dpi | w/ (50) | × | | | 16.40 | 5.50 | - | - |
| | Voigtlaender et al. (2016) | GPU-based, diagonal MDLSTM | w/o | | | | 9.30 | 3.50 | 9.60 | 2.80 |
| | SepMDLSTM (Chen et al., 2017) | Multi-task approach | w/o | | | | 34.55 | 11.15 | 30.54 | 8.29 |
| | Bluche et al. (2017) | MDLSTM, attention | w/o | × | | | - | 16,20 | - | - |
| | | Line segmentation 150 dpi | w/o | | × | | - | 11.10 | - | - |
| | | Line segmentation 150 dpi | w/o | | × | | - | 7.50 | - | - |
| | MDLSTM (Castro et al., 2018) | | | | | | 10.50 | 3.60 | - | - |
| **RNN** | BiLSTM (Graves et al., 2009) | | w/ (20) | | | | 18.20 | 25.90 | - | - |
| | HMM+RNN (Menasri et al., 2012) | Sliding win. Gaussian HMM, RNN | | | × | × | | - | 4.75 | - |
| | Dropout (Pham et al., 2014) | LSTMs with dropout | w/o | | | | 35.10 | 10.80 | 28.50 | 6.80 |
| | Voigtlaender et al. (2015) | Maximum mutual information | | | | | 12.70 | 4.80 | 12.10 | 4.40 |
| | Bluche (2015) | | | | | | 10.90 | 4.40 | 11.20 | 3.50 |
| | | | w/ (50) | | | | 13.60 | 5.10 | 12.30 | 3.30 |
| | GCRNN (Bluche & Messina, 2017) | CNN+BiLSTM | w/ (50) | | | | 10.50 | 3.20 | 7.90 | 1.90 |
| | CNN-1DLSTM-CTC (Puigcerver, 2017) | CNN+BiLSTM+CTC (128 × width) | w/o | | × | | 18.40 | 5.80 | 9.60 | 2.30 |
| | | NN+BiLSTM+CTC | w/ (50) | | × | | 12.20 | 4.40 | 9.00 | 2.50 |
| | End2End (Krishnan et al., 2018) | Without line level | w/ | | | | 16.19 | 6.34 | - | - |
| | | Line level | w/ | | × | | 32.89 | 9.78 | - | - |
| | SFR (Wigington et al., 2018) | Text detection and segmentation | w/o | × | | | 23.20 | 6.40 | 9.30 | 2.10 |
| | CNN-RNN (Dutta et al., 2018) | Unconstrained | w/o | | | | 12.61 | 4.88 | 7.04 | 2.32 |
| | | Full-Lexicon | w/ | | | | 4.80 | 2.52 | 1.86 | 0.65 |
| | | Text-Lexicon | w/ | | | | 4.07 | 2.17 | | |
| | | Unconstrained | w/o | | × | | 17.82 | 5.70 | 9.60 | 2.30 |
| | Chowdhury & Vig (2018) | Seq2seq, w/o LN | w/o | | | | 25.50 | 17.40 | 19.10 | 12.00 |
| | | w/ LN | w/o | | | | 22.90 | 13.10 | 15.80 | 9.70 |
| | | w/ LN + Focal Loss | w/o | | | | 21.10 | 11.40 | 13.50 | 7.30 |
| | | w/ LN + Focal Loss + Beam Search | w/o | | | | 16.70 | 8.10 | 9.60 | 3.50 |
| | Sueiras et al. (2018) | LSTM encoder-decoder, attention | | | | | 15.90 | 4.80 | - | - |
| | Chung & Delteil (2019) | ResNet+LSTM, segmentation | w/ | × | | | - | 8.50 | - | - |
| | Ingle et al. (2019) | BiLSTM | | | × | | 30.70 | 12.80 | - | - |
| | | GRCL | | | × | | 35.20 | 14.10 | - | - |
| | Michael et al. (2019) | Seq2seq CNN+BiLSTM (64 × width) | | | × | | - | 5.24 | - | - |
| | FPN (Carbonell et al., 2019) | Feature Pyramid Network, 150 dpi | | | × | | - | 15.60 | - | - |
| | AFDM (Bhunia et al., 2019) | AFD module | w/ | | | | 8.87 | 5.94 | 6.31 | 3.17 |
| **CNN** | Poznanski & Wolf (2016) | CNN + connected branches, CCA | w/ | | | | 6.45 | 3.44 | 3.90 | 1.90 |
| | Gated text recognizer (Yousef et al., 2020) | CNN+CTC (32 × width) | w/o | | × | | - | 4.90 | - | - |
| | OrigamiNet (Yousef & Bishop, 2020) | VGG (500 × 500) | × | × | | | - | 51.37 | - | - |
| | | VGG (500 × 500), w/o LN | w/o | × | × | | - | 34.55 | - | - |
| | | ResNet26 (500 × 500), w/o LN | w/o | × | × | | - | 10.03 | - | - |
| | | ResNet26 (500 × 500), w/ LN | w/o | × | × | | - | 7.24 | - | - |
| | | ResNet26 (500 × 500), w/o LN | w/o | × | × | | - | 8.93 | - | - |
| | | ResNet26 (500 × 500), w/ LN | w/o | × | × | | - | 6.37 | - | - |
| | | ResNet26 (500 × 500), w/o LN | w/o | × | × | | - | 76.90 | - | - |
| | | ResNet26 (500 × 500), w/ LN | w/o | × | × | | - | 6.13 | - | - |
| | | GTR-8 (500 × 500), w/o LN | w/o | × | × | | - | 72.40 | - | - |
| | | GTR-8 (500 × 500), w/ LN | w/o | × | × | | - | 5.64 | - | - |
| | | GTR-8 (750 × 750), w/ LN | w/o | × | × | | - | 5.50 | - | - |
| | | GTR-12 (750 × 750), w/ LN | w/o | × | × | | - | 4.70 | - | - |
| | DAN (Wang et al., 2020b) | Decoupled attention module | w/o | | × | | 19.60 | 6.40 | 8.90 | 2.70 |
| **GAN** | ScrabbleGAN (Fogel et al., 2020) | Original data | w/o | | | | 25.10 | - | 12.29 | - |
| | | Augmentation | w/o | | | | 24.73 | - | 12.24 | - |
| | | Augmentation + 100k synth. | w/o | | | | 23.98 | - | 11.68 | - |
| | | Augmentation + 100k synth. + Refine | w/o | | | | 23.61 | - | 11.32 | - |
| **Trans-former** | Kang et al. (2022) | Self-attention for text/images | w/o | | × | | 15.45 | 4.67 | - | - |
| | FPHR (Singh & Karayev, 2021) | CNN encoder, Transformer decoder | w/o | × | | | - | 6.70 | - | - |
| | | With augmentation | w/o | × | | | - | 6.30 | - | - |
| **Other** | FST (Messina & Kermorvant, 2014) | Finite state transducer (lexicon) | n-gram | | | | 19.10 | - | 13.30 | - |

**Abbreviations.** Size $k$ of the language model (LM), i.e., with (w/) or without (w/o) a LM. P: paragraph or full text level, L: line level, LN: layer normalization, CER: character error rate, WER: word error rate, MLP: multi-layer perceptron, HMM: hidden markov model, GTR: gated text recognizer, seq2seq: sequence-to-sequence, GAN: generative adversarial network, CTC: connectionist temporal classification RNN: recurrent neural network, LSTM: long short-term memory, CNN: convolutional neural network

slant correction. The methods by Chowdhury & Vig (2018); Sueiras et al. (2018); Ingle et al. (2019); Michael et al. (2019) also make use of BiLSTMs. While Carbonell et al. (2019) uses a feature pyramid network (FPN), the adversarial feature deformation module (AFDM) (Bhunia et al., 2019) learns ways to elastically warp extracted features in a scalable manner. Further methods that combine CNNs with RNNs are (Liang et al., 2017; Sudholt & Fink, 2018; Xiao & Cho, 2016), while BiLSTMs are utilized in (Carbune et al., 2020; Tian et al., 2019).

**TCNs.** TCNs use dilated causal convolutions and have been applied to air-writing recognition by Bastas et al. (2020). As RNNs are slow to train, Sharma et al. (2020) presented a faster system that is based on text line images and TCNs with the CTC loss. This method achieves 9.6% CER on the IAM-OffDB dataset. Sharma & Jayagopi (2021) combined 2D convolutions with 1D dilated non-causal convolutions that offer high parallelism with a smaller number of parameters. They analyzed re-scaling factors and data augmentation and achieved comparable results for the IAM-OffDB and RIMES datasets.

**CNNs.** Poznanski & Wolf (2016) utilized a CNN with multiple fully connected branches to estimate its n-gram frequency profile (set of n-grams contained in the word). With canonical correlation analysis (CCA), the estimated profile can be matched to the true profiles of all words in a large dictionary. As most attention methods suffer from an alignment problem, Wang et al. (2020b) proposed a decoupled attention network (DAN) that has a convolutional alignment module that decouples the alignment operation from using historical decoding results based on visual features. The gated text recognizer (Yousef et al., 2020) aims to automate the feature extraction from raw input signals with a minimum required domain knowledge. The fully convolutional network without recurrent connections is trained with the CTC loss. Thus, the gated text recognizer module can handle arbitrary input sizes and can recognize strings with arbitrary lengths. This module has been used for OrigamiNet (Yousef & Bishop, 2020) which is a segmentation-free multi-line or full-page recognition system. OrigamiNet yields state-of-the-art results on the IAM-OffDB dataset, and shows improved performance of gated text recognizer over VGG and ResNet26. Hence, we use the gated text recognizer module as our visual feature encoder for offline HWR (see Section A.5).

**GANs.** Handwriting text generation is a relatively new field. The first approach by Graves (2014) was a method to synthesize online data based on RNNs. The technique HWGAN by Ji & Chen (2020) extends this method by adding a discriminator $\mathcal{D}$. DeepWriting (Aksan et al., 2018) is a GAN that is capable of disentangling style from content and thus making digital ink editable. Haines et al. (2016) proposed a method to generate handwriting based on a specific author with learned parameters for spacing, pressure, and line thickness. Alonso et al. (2019) used a BiLSTM to obtain an embedding of the word to be rendered and added an auxiliary network as a recognizer $\mathcal{R}$. The model is trained with a combination of an adversarial loss and the CTC loss. ScrabbleGAN by Fogel et al. (2020) is a semi-supervised approach that can arbitrarily generate many images of words with arbitrary length from a generator $\mathcal{G}$ to augment handwriting data and uses a discriminator $\mathcal{D}$ and recognizer $\mathcal{R}$. The paper proposes results for original data with random affine augmentation using synthetic images and refinement.

**Transformers.** RNNs prevent parallelization, due to their sequential pipelines. Kang et al. (2022) introduced a non-recurrent model by the use of Transformer models with multi-head self-attention layers at the textual and visual stages. Their method works for any pre-defined vocabulary. For the feature encoder, they used modified ResNet50 models. The full page HTR (FPHR) method by Singh & Karayev (2021) uses a CNN as an encoder and a Transformer as a decoder with positional encoding.

## A.2 Overview of Cross-Modal Retrieval Methods

We provide a summary of methods for cross-modal learning in Table 10 and Table 11. Typical modalities are video, image, audio, text, sensors (such as inertial sensors used for our method), and haptic modalities. We classify each method with the technique used for pairwise learning that utilizes an objective for deep metric learning. The overview contains a wide range of applications, while visual semantic embedding is a common field for cross-modal retrieval.

Table 10: Overview of cross-modal and pairwise learning techniques using the modalities video (V), images (I), audio (A), text (T), sensors (S), or haptic (H). Data from sensors are represented by time-series from inertial, biological, or environmental sensors. We indicate cross-modal learning from the same modality with $\times^n$ with $n$ modalities. If $n$ is unspecified, the method can potentially work with an arbitrary number of modalities.

| Method (sorted by year) | Modality | | | | | | Pairwise Learning | Deep Metric Learning Loss/Objective | Application |
|---|---|---|---|---|---|---|---|---|---|
| | V | I | A | T | S | H | | | |
| Chopra et al. (2005) | | $\times^2$ | | | | | Pairwise | $L_1$ similarity | Face verification |
| Rasiwasia et al. (2010) | | $\times$ | | $\times$ | | | Pairwise | Canonical correlation analysis | Multimedia documents: emb. mapping to common space |
| DeViSE (Frome et al., 2013) | | $\times$ | | $\times$ | | | Hinge rank | Cosine similarity | Visual semantic embedding |
| OxfordNet (Kiros et al., 2014) | | $\times$ | | $\times$ | | | Contrastive | Cosine similarity | Visual semantic embedding |
| Young et al. (2014) | | $\times$ | | $\times$ | | | Denotion graph | Pointwise MI | Visual semantic embedding |
| DAN (Long et al., 2015) | | $\times^2$ | | | | | Pairwise | Kernelized MMD | Domain adaptation |
| ml-CCA (Ranjan et al., 2015) | | $\times$ | | $\times$ | | | Not pairwise | CCA extended | Multi-label annotations |
| FaceNet (Schroff et al., 2015) | | $\times$ | | | | | Triplet | Euclidean | Face recognition, clustering |
| deep-SM (Wei et al., 2016) | | $\times^2$ | | $\times$ | | | Pairwise | CCA, T-V CCA semantic matching | Universal representation for various recognition tasks |
| Gordo & Larlus (2017) | | $\times$ | | $\times$ | | | Triplet | non-Mercer match kernel | Visual semantic embedding |
| TristouNet (Bredin, 2017) | | | $\times$ | | | | Triplet | Euclidean | Speech classification |
| Triplet+FANNG (Harwood et al., 2017) | | $\times$ | | | | | Smart triplet | Nearest neighbour graph | General |
| Zeng et al. (2017) | | $\times$ | | | | | Triplet | CE, conditional random field | Handwritten Chinese characters recognition |
| Huang et al. (2018) | | $\times$ | | $\times$ | | | Pairwise | Cosine similarity | Visual semantic embedding |
| GXN (Gu et al., 2018) | | $\times$ | | $\times$ | | | Triplet | Similarity: order-violation penalty | Visual semantic embedding |
| TDH (Deng et al., 2018) | | $\times^2$ | | $\times$ | | | Triplet | Hamming space | Visual semantic embedding |
| VSE++ (Faghri et al., 2018) | | $\times$ | | $\times$ | | | Triplet | Similarity: inner product | Visual semantic embedding |
| SCAN t-i (Lee et al., 2018) | | $\times$ | | $\times$ | | | Triplet | Similarity LSE | Visual semantic embedding |
| Discriminative (Do et al., 2019) | | $\times$ | | | | | Triplet | Class centroids | Image classification |
| VSRN (Li et al., 2019) | | $\times$ | | $\times$ | | | Triplet | Similarity: inner product | Visual semantic embedding |
| PIE-Nets (Song & Soleymani, 2019) | $\times$ | $\times$ | | $\times$ | | | Pairwise | Diversity, MIL, MMD | Visual semantic embedding |
| LIWE (Wehrmann et al., 2019) | | $\times$ | | $\times$ | | | Contrastive | Sum/Max of Hinges | Visual semantic embedding |
| Zhang et al. (2019) | | $\times$ | | | | | STriplet+triplet | Cosine similarity | Relationship understanding |
| TIMAM (Sarafianos et al., 2019) | | $\times$ | | $\times$ | | | Pairwise | Norm-softmax CE | Visual question answering |
| GMN (Lim et al., 2019) | | $\times^n$ | $\times^n$ | | | $\times^n$ | Pairwise | Cross-modal generation | Multisensory 3D scenes |
| CTM (Guo et al., 2019) | $\times$ | | | $\times$ | | | Triplet | CTC, CE , $L_2$ correlation | Sentence translation |
| UniVSE (Wu et al., 2019) | | $\times$ | | $\times$ | | | Contrastive | Alignment losses | Visual semantic embedding |
| ActiveSet+RRPB (Yoshida et al., 2019) | | $\times$ | | | | | Smart triplet | Semidefinite constraint | General |
| PAN (Zhen et al., 2019) | | $\times$ | | $\times$ | | | Pairwise | Cosine similarity | Visual semantic embedding |
| CM-GANs (Peng & Qi, 2019) | | $\times$ | | $\times$ | | | Adversarial | Inter/intra class | Visual semantic embedding |
| CPC (van den Oord et al., 2019) | | $\times$ | $\times$ | | $\times$ | | Contrastive | CE, MI | One modality classification |
| CrossATNet (Chaudhuri et al., 2020) | | $\times^2$ | | $\times$ | | | Triplet | MSE | Zero-shot learning, sketches |
| MHTN (Huang et al., 2020) | $\times$ | $\times$ | $\times$ | $\times$ | | | Pairwise, contr. | MMD, Euclidean | CMR |
| GCML (Lee et al., 2020) | $\times^2$ | | $\times$ | | | | Triplet | Hierarchical relational graph clustering | Retrieval, search, video-to-video similarity |
| CSVE (Wang et al., 2020a) | | $\times$ | | $\times$ | | | Bidirect. triplet | Correlation graph | Visual semantic embedding |
| TXS-Adapt (Semedo & Magalhães, 2020) | | $\times$ | | $\times$ | | | Triplet (adaptive) | Recency-based correlation | Social media domain |
| Proxy-Anchor (Kim et al., 2020) | | $\times$ | | | | | Pair+proxy | Cosine similarity | Image classification |
| TNN-C-CCA (Zeng et al., 2020) | $\times$ | | $\times$ | | | | Triplet | CCA | Multimedia |
| AdapOffQuin (Chen et al., 2020) | | $\times$ | | $\times$ | | | Quintuplet | Cosine similarity | Visual semantic embedding |
| ROMA (Li et al., 2021) | | $\times^2$ | | | | | Soft triplet (fixed margin) | CE, random perturbation | Unsupervised representation learning |
| CLIP (Radford et al., 2021) | | $\times$ | | $\times$ | | | Contrastive | CE, Cosine similarity | OCR, action/object recognition |
| SGRAF (Diao et al., 2021) | | $\times$ | | $\times$ | | | Pairwise | Vector similarity | Visual semantic embedding |
| PCME (Chun et al., 2021) | | $\times$ | | $\times$ | | | Triplet | Euclidean | Visual semantic embedding |
| MCN (Chen et al., 2021a) | $\times$ | | $\times$ | $\times$ | | | Contrastive | Similarity, reconstruction | Multimodal clustering |
| VATT (Akbari et al., 2021) | $\times$ | | $\times$ | $\times$ | | | Contrastive | CC, NCE, MIL-NCE | Transformer for CMR |
| Wan & Zou (2021) | | $\times$ | | | | | Dual triplet | Euclidean | Signature verification |
| Wang et al. (2021) | $\times$ | | $\times^2$ | | | | Contrastive | Cosine similarity | Audio classification |

Table 11: Table 10 continued.

| Method (sorted by year) | Modality V | I | A | T | S | H | Pairwise Learning | Deep Metric Learning Loss/Objective | Application |
|---|---|---|---|---|---|---|---|---|---|
| Hafner et al. (2022) | | $\times^2$ | | | | | Triplet | Softmax, MSE | Person re-identification |
| AlignMixup (Venkataramanan et al., 2022) | | $\times^2$ | | | | | Pairwise | Sinkhorn transport | Data augmentation for interpolation |
| SAM (Biten et al., 2022) | | $\times$ | | $\times$ | | | Triplet | Cosine similarity | Visual semantic embedding |
| VSE$_\infty$ (Chen et al., 2022b) | | $\times$ | | $\times$ | | | Triplet | Similarity | Visual semantic embedding |
| AudioCLIP (Guzhov et al., 2022) | | $\times$ | $\times$ | $\times$ | | | Contrastive | Cosine similarity, symmetric CE | Environmental sound classification |
| data2vec (Baevski et al., 2022) | $\times$ | | $\times$ | $\times$ | | | Predicts latent representations | | Self-supervision with masks |
| ColloSSL (Jain et al., 2022) | | | | | $\times^n$ | | Contrastive | CE, Cosine similarity | Human-activity recognition |
| COCOA (Deldari et al., 2022b) | | | | | $\times^n$ | | Contrastive | Cosine similarity | General time-series |
| ELo (Piergiovanni et al., 2022) | $\times$ | $\times$ | $\times$ | $\times$ | | | Contrastive | $L_2$, evolutionary | Cross-modal, multi-task |
| Lin et al. (2022) | | $\times$ | | | | | Pairwise | - | Crowd counting |
| MM-ALT (Gu et al., 2022) | $\times$ | | $\times$ | | $\times$ | | Pairwise | CTC, residual attention | Lyric transcription |
| FLAVA (Singh et al., 2022) | | $\times$ | | $\times$ | | | Contrastive | Cosine similarity, temperature scaling | Visual semantic embedding |
| ConceptBeam (Ohishi et al., 2022) | | $\times$ | $\times^n$ | | | | Triplet | Cosine similarity | Target speech extraction |
| Falcon et al. (2022) | $\times$ | | | $\times$ | | | Contrastive | - | Text-video retrieval, kitchen |
| C$^3$CMR (Wang et al., 2022) | | $\times$ | | $\times$ | | | Triplet | CE, cosine similarity | Visual semantic embedding |
| Chen et al. (2022a) | | $\times$ | | $\times$ | | | Contrastive | Cosine similarity | VSE with graph embedding |
| Ott et al. (2022a) | | | | $\times^2$ | | | Classwise | CE, HoMM/CC/PC | Online HWR |
| **CMR-IS (Ours, 2022)** | | $\times$ | | $\times$ | | | Contr., triplet | CTC, MSE/CC/PC/KL | Online HWR |

**Abbreviations.** CE: cross-entropy, CTC: connectionist temporal classification, MSE: mean squared error, CC: cross-correlation, PC: Pearson correlation, MMD: maximum mean discrepancy, HoMM: higher-order moment matching, CCA: canonical correlation analysis, MIL: multiple-instance learning, MI: mutual information, NCE: noise contrastive estimation, VSE: visual semantic embedding

## A.3 Multi-Task Learning

We simultaneously train the $\mathcal{L}_{\text{CTC}}$ loss for sequence classification combined with one or two shared losses $\mathcal{L}_{\text{shared},1}$ and $\mathcal{L}_{\text{shared},2}$ for cross-modal representation learning. As both losses are in different ranges, the naive weighting

$$\mathcal{L}_{\text{total}} = \sum_{i=1}^{|T|} \omega_i \mathcal{L}_i, \tag{3}$$

with pre-specified constant weights $\omega_i = 1, \forall i \in \{1, \ldots, |T|\}$ can harm the training process. Hence, we apply dynamic weight average (DWA) (Liu et al., 2019) as a multi-task learning approach that performs dynamic task weighting over time (i.e., after each batch).

## A.4 Training Synthetic Data with the Triplet Loss

**Signal and Image Generation.** We combine the networks for both signal and image classification to improve the classification accuracy over each single-modal network. The aim is to show that the triplet loss can be used for such a cross-modal setting in the field of cross-modal representation learning. Hence, we generate synthetic data in which the image data contains information of the signal data. We generate signal data $\mathbf{x}$ with $x_{i,k} = \sin\left(0.05 \cdot \frac{t_i}{k}\right)$ for all $t_i \in \{1, \ldots, 1,000\}$ where $t_i$ is the timestep of the signal. The frequency of the signal is dependent on the class label $k$. We generate signal data for 10 classes (see Figure 10a). We add noise from a continuous uniform distribution $U(a, b)$ for $a = 0$ and $b = 0.3$ (see Figure 10b) and add time and magnitude warping (see Figure 10c). We generate a signal-image pair such that the image is based on the signal data. We make use of the Gramian angular field that transforms time-series into images. The time-series is defined as $\mathbf{x} = (x_1, \ldots, x_n)$ for $n = 1,000$. The Gramian angular field creates a matrix of temporal correlations for each $(x_i, x_j)$ by rescaling the time-series in the range $[p, q]$ with $-1 \le p < q \le 1$ by

$$\hat{x}_i = p + (q - p) \cdot \frac{x_i - \min(\mathbf{x})}{\max(\mathbf{x}) - \min(\mathbf{x})}, \forall i \in \{1, \ldots, n\}, \tag{4}$$

and computes the cosine of the sum of the angles for the Gramian angular summation field (Wang & Oates, 2015) by

$$\text{GASF}_{i,j} = \cos\left(\phi_i + \phi_j\right), \forall i, j \in 1, \ldots, n, \tag{5}$$

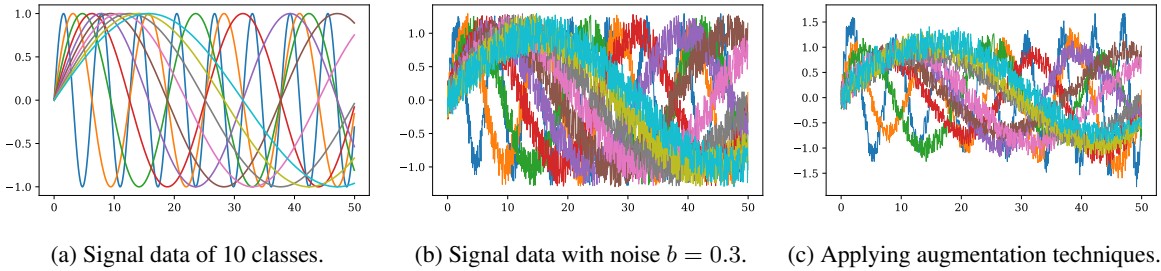

(a) Signal data of 10 classes.        (b) Signal data with noise $b = 0.3$.        (c) Applying augmentation techniques.

Figure 10: Plot of the 1D signal data for 10 classes.

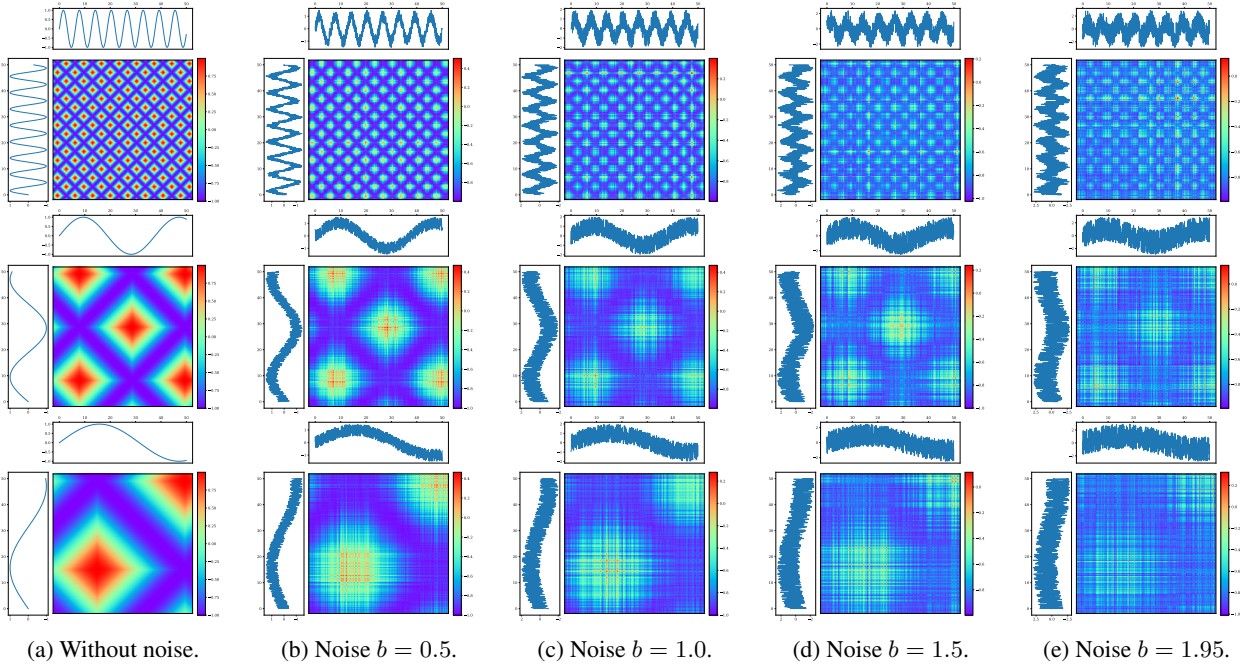

(a) Without noise.        (b) Noise $b = 0.5$.        (c) Noise $b = 1.0$.        (d) Noise $b = 1.5$.        (e) Noise $b = 1.95$.

Figure 11: Plot of the Gramian angular summation field based on 1D signal data with added noise for the classes 0 (top row), 5 (middle row) and 9 (botton row).

with $\phi_i = \arccos(\hat{x}_i), \forall i \in \{1, \ldots, n\}$ being the polar coordinates. We generate image datasets based on signal data with different noise parameters ($b \in \{0.0, \ldots, 1.95\}$) to show the influence of the image data on the classification accuracy. As an example, Figure 11 shows the Gramian angular summation field plots for the noise parameters $b = [0, 0.5, 1.0, 1.5, 1.95]$. We present the Gramian angular summation field for the classes 0, 5, and 9 to show the dependency of the frequency of the signal data on the Gramian angular summation field.

**Models.** We use the following models for classification. Our encoder for time-series classification consists of a 1D convolutional layer (filter size 50, kernel 4), a max pooling layer (pool size 4), batch normalization, and a dropout layer (20%). The image encoder consists of a layer normalization and 2D convolutional layer (filter size 200), and batch normalization with ELU activation. After that, we add a 1D convolutional layer (filter size 200, kernel 4), max pooling (pool size 2), batch normalization, and 20% dropout. For both models, after the dropout layer follows a cross-modal representation – i.e., an LSTM with 10 units, a Dense layer with 20 units, a batch normalization layer, and a Dense layer of 10 units (for 10 sinusoidal classes). These layers are shared between both models.

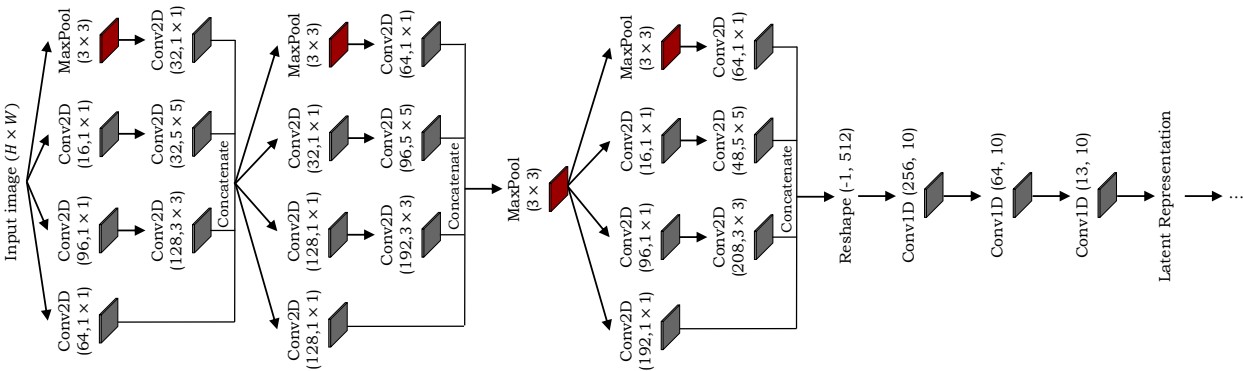

Figure 12: Offline HWR method based on Inception modules (Szegedy et al., 2015).

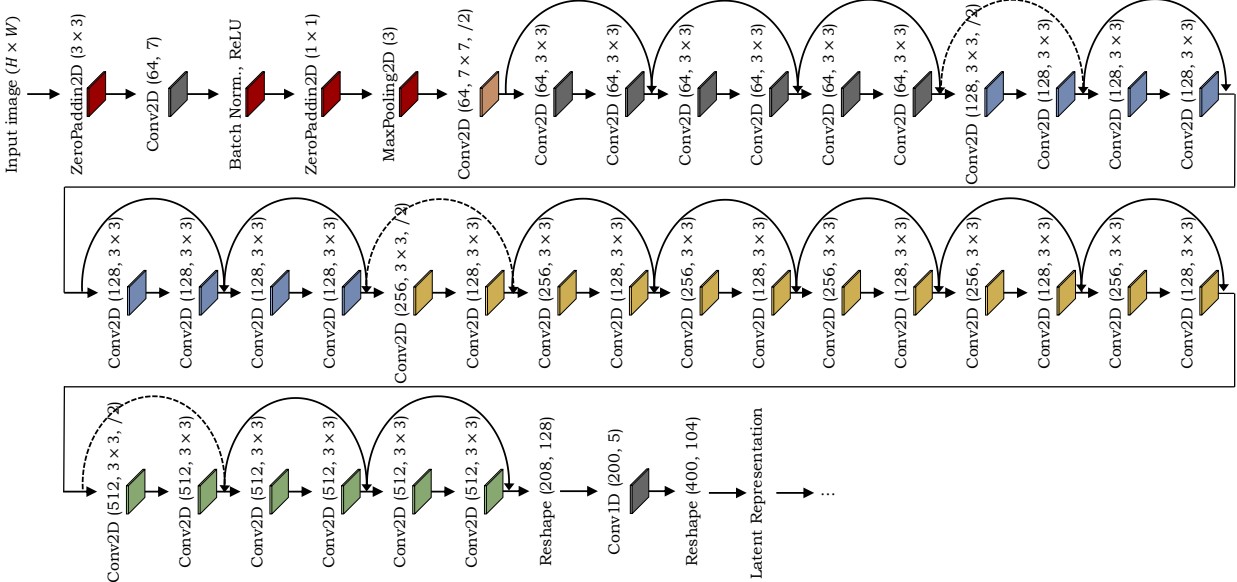

Figure 13: Offline HWR method based on the ResNet34 architecture (He et al., 2016).

## A.5  Details on Architectures for Offline HWR

In this section, we provide details about the integration of `Inception` (Szegedy et al., 2015), `ResNet` (He et al., 2016) and `gated text recognizer` (Yousef et al., 2020) modules into the offline HWR system. All three architectures are based on publicly available implementations, but we changed or adapted the first layer for the image input and the last layer for a proper input for our latent representation module.

**Inception.**  Figure 12 gives an overview of the integration of the Inception module. The Inception module is part of the well-known GoogLeNet architecture. The main idea is to consider how an optimal local sparse structure can be approximated by readily available dense components. As the merging of pooling layer outputs with convolutional layer outputs would lead to an inevitable increase in the number of output and would lead to a high computational increase, we apply the Inception module with dimensionality reduction to our offline HWR approach (Szegedy et al., 2015). The input image is of size $H \times W$. What follows is the Inception (3a), Inception (3b), a max pooling layer $(3 \times 3)$ and Inception (4a). We add three 1D convolutional layers to obtain an output dimensionality of $400 \times 200$ as the input for the latent representation.

**ResNet34.**  Figure 13 provides an overview of the integration of the ResNet34 architecture. Instead of learning unreferenced functions, He et al. (2016) reformulated the layers as learning residual functions with reference to the

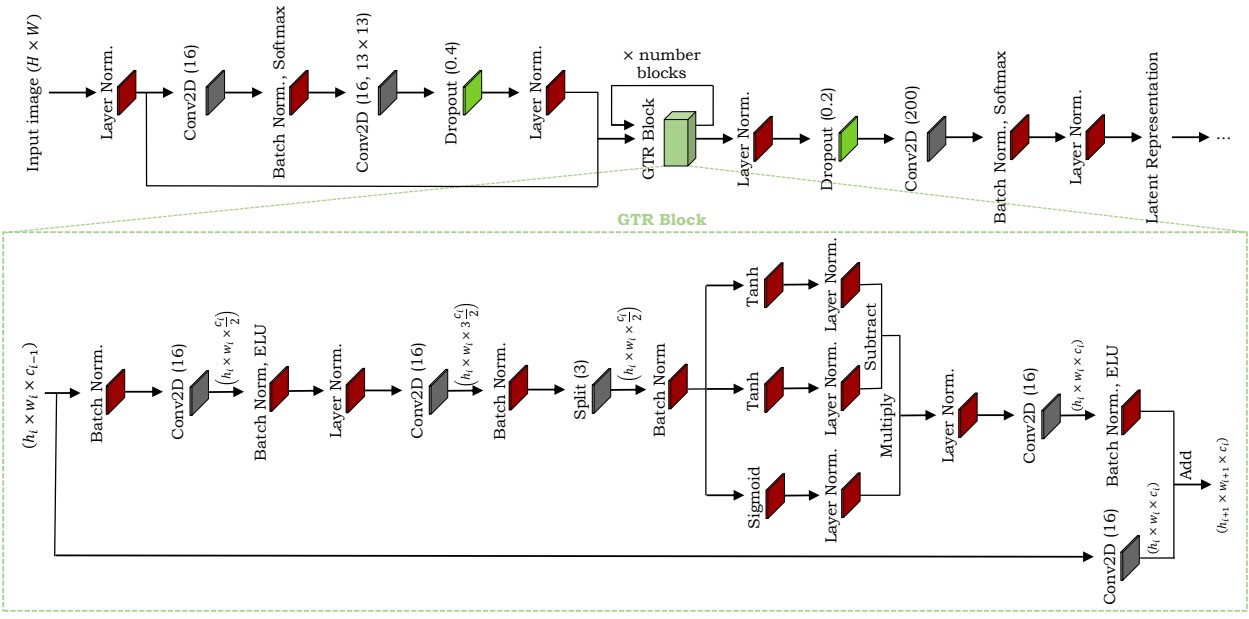

Figure 14: Offline HWR method based on the gated text recognizer architecture (Yousef et al., 2020).

layer inputs. This residual network is easier to optimize and can gain accuracy from considerably increased depth. The ResNet block allows the layers to fit a residual mapping denoted as $\mathcal{H}(\mathbf{x})$ with identity $\mathbf{x}$ and fits the mapping $\mathcal{F}(\mathbf{x}) := \mathcal{H}(\mathbf{x}) - \mathbf{x}$. The original mapping is recast into $\mathcal{F}(\mathbf{x}) + \mathbf{x}$. We reshape the output of ResNet34, add a 1D convolutional layer, and reshape the output for the latent representation.

**Gated Text Recognizer.** Figure 14 gives an overview of the integration of the gated text recognizer (Yousef et al., 2020) module – a fully convolutional network that uses batch normalization and layer normalization to regularize the training process and increase convergence speed. The module uses batch renormalization (Ioffe, 2017) on all batch normalization layers. Depthwise separable convolutions reduce the number of parameters at the same/better classification performance. The gated text recognizer uses spatial dropout instead of regular unstructured dropout for better regularization. After the input image of size $H \times W$ that is normalized follows a convolutional layer with Softmax normlization, a $13 \times 13$ filter, and dropout (40%). After the dropout layer, a stack of 2, 4, 6 or 8 gate blocks follows that models the input sequence. Similar to (Yousef et al., 2020), we add a dropout of 20% after the last gated text recognizer block. Lastly, we add a 2D convolutional layer of 200, a batch normalization layer and a layer normalization layer that is the input for our latent representation.

## A.6 Detailed Online HWR Evaluation

Table 7 gives an overview of cross-modal representation learning results based on two convolutional layers ($c = 2$) for the cross-modal representation. Our CNN+BiLSTM contains three additional convolutional layers and outperforms the smaller CNN+BiLSTM by (Ott et al., 2022c) on the WD classification tasks. Without triplet loss, $\mathcal{L}_{PC}$ yields the best results on the OnHW-wordsRandom dataset. The triplet loss partly decreases results and partly improves results on the OnHW-words500 dataset. In conclusion, two convolutional layers for the cross-modal representation has a negative impact, while here the triplet loss has no impact.

Table 12: Evaluation results (WER and CER in %) averaged over five splits of the baseline time-series-only technique and our cross-modal learning technique for the inertial-based OnHW datasets (Ott et al., 2022c) with and without mutated vowels (MV) for two convolutional layers $c = 2$. We propose writer-(in)dependent (WD/WI) results. Best results are **bold**, and second best results are underlined. Arrows indicate improvements (↑) and degradation (↓) of baseline results (CNN+BiLSTM, w/o MV).

| | OnHW-words500 | | | | OnHW-wordsRandom | | | |
| | WD | | WI | | WD | | WI | |
| **Method** | **WER** | **CER** | **WER** | **CER** | **WER** | **CER** | **WER** | **CER** |
| Small CNN+BiLSTM, $\mathcal{L}_{\text{CTC}}$, w/ MV | 51.95 | 17.16 | 60.91 | 27.80 | 41.27 | 7.87 | 84.52 | 35.22 |
| CNN+BiLSTM (ours), $\mathcal{L}_{\text{CTC}}$, w/ MV | 42.81 | 13.04 | 60.47 | 28.30 | 37.13 | 6.75 | 83.28 | 35.90 |
| CNN+BiLSTM (ours), $\mathcal{L}_{\text{CTC}}$, w/o MV | 42.77 | 13.44 | 59.82 | 28.54 | 41.52 | 7.81 | 83.54 | 36.51 |
| $\mathcal{L}_{\text{MSE}}$ | 39.79 ↑ | 12.14 ↑ | 60.35 ↓ | 28.48 ↑ | 39.98 ↑ | 7.79 ↑ | 83.50 ↑ | 36.92 ↓ |
| $\mathcal{L}_{\text{CS}}$ | 43.40 ↓ | 13.70 ↓ | 59.31 ↑ | 27.99 ↑ | 40.31 ↑ | 7.68 ↑ | 83.68 ↓ | 36.30 ↑ |
| $\mathcal{L}_{\text{PC}}$ | 38.90 ↑ | 11.60 ↑ | 60.77 ↓ | 28.45 ↑ | **39.93** ↑ | **7.60** ↑ | **83.19** ↑ | **35.83** ↑ |
| $\mathcal{L}_{\text{KL}}$ | **37.25** ↑ | **11.29** ↑ | 65.10 ↓ | 31.26 ↓ | 41.81 ↓ | 8.22 ↓ | 84.40 ↓ | 38.93 ↓ |
| $\mathcal{L}_{\text{trpl,2}}(\mathcal{L}_{\text{MSE}})$ | 41.16 ↑ | 12.71 ↑ | 58.65 ↑ | 28.19 ↑ | 41.16 ↑ | 8.03 ↓ | 85.38 ↓ | 39.49 ↓ |
| $\mathcal{L}_{\text{trpl,2}}(\mathcal{L}_{\text{CS}})$ | 42.74 ↑ | 13.43 ↑ | **58.13** ↑ | **27.62** ↑ | 41.49 ↑ | 8.18 ↓ | 85.24 ↓ | 38.75 ↓ |
| $\mathcal{L}_{\text{trpl,2}}(\mathcal{L}_{\text{PC}})$ | 39.94 ↑ | 12.19 ↑ | 62.76 ↓ | 30.68 ↓ | 41.58 ↓ | 8.18 ↓ | 85.18 ↓ | 38.53 ↓ |
| $\mathcal{L}_{\text{trpl,2}}(\mathcal{L}_{\text{KL}})$ | 38.34 ↑ | 11.77 ↑ | 67.08 ↓ | 33.84 ↓ | 41.87 ↓ | 8.33 ↓ | 86.34 ↓ | 40.37 ↓ |

