# OpenReview forum: "Auxiliary Cross-Modal Representation Learning with Triplet Loss Functions for Online Handwriting Recognition"
_TMLR — Rejected by TMLR_

### Review · Reviewer_DLNN · 2022-10-18

**Summary Of Contributions:**

In this paper, the authors proposed a new framework for online handwriting recognition with cross-modal representation learning between time-series data and images.
1. The authors successfully generated paired signal and image data for handwriting for pretraining.
2. A model inspired by recent work sharing encoders among modalities like VATT is proposed for image-signal representation learning.

**Audience:**

Yes

**Broader Impact Concerns:**

How does this representation work for people with certain disabilities?

**Claims And Evidence:**

Yes

**Requested Changes:**

Please check cons for details.

**Strengths And Weaknesses:**

Pros:
1. The proposed framework is new and intuitive.
2. The empirical results are strong.

Cons:
1. Although the overall framework is new and interesting, each component of the proposed model is based on existing research, which limits the technical innovation of the work.
2. The presentation of the paper really needs to be improved.
a. Figure 1 and Figure didn't clearly specify the training and testing settings. It is somewhat confusing that the audience may not be able to understand which part of the network is used in pretraining, finetuning, and inference, respectively.
b. The Tables summarize the comparison with existing works is nice for experienced researchers in this topic but not really friendly for audience who are not that familiar without more understanding of the definition of the tasks and terminologies.
c. The approach part could also use better organization. Details like network layers could be moved to implementation details. The whole pipeline lacks more fornal mathematical descriptions for each component.
3. It is still unclear why using images as training data improves the performance.
a. Figure 9 a) and b) are hard to understand without labels for each clusters/points. It is also hard to really make any conclusion on which one is better without further explanations.
b. Conceptually, the images could only provide more noisy gradients than directly using the condition(label) for generating them. But why such noisier gradients may provide better initialization for the mode? More comparison and justification should be provided. For example, what if simply add more noisy in the signal synthetic process and use it for pretraining only.
c. Does the learned representation without finetuning generalize well? Linear-probing/kNN analysis is interesting and necessary to see.
4. This is more like an open discussion. How to compare this model with exiting products that have already worked very well, like the function in iPad OS?

---

> ### Author Response · Authors · 2022-10-22
> **Response to Reviewer DLNN**
>
> We would like to thank the reviewer for the thoughtful comments and efforts towards improving our manuscript. In the following, we provide a response to the raised questions by the reviewer:
>
> 1. The reviewer is correct that components of our method are based on existing research (i.e., OnHW classification model, image generation with ScrabbleGAN, OCR classification with GTR modules, and cross-modal learning with pairwise/triplet learning). However, we would like to point out that cross-modal learning between images and time-series data is rare (see Table 2), and a dynamic margin for the triplet loss based on the Edit distance is novel. Furthermore, our approach shows that the recent methods ScrabbleGAN (2020) and OrigamiNet (2020) are applicable in the real-world setup of offline HWR to enhance the online HWR task. Hence, our work both proposes  methodological advancements and also transfers existing ideas to a new domain. We make this now more clear in the paragraph “Our Contribution”.
>
> 2. a) We understand that some training details are missing in Figure 1 and Figure 4, i.e., highlighting the part of pre-training, fine-tuning, and inference. We will revise the contents accordingly after having received all other reviews.
> b) We introduced the two overview tables for handwriting recognition and cross-modal learning, as our method covers both aspects and there exists much research in both fields. We defined the terminologies required for our method in Section 3.1 for the notation of the multivariate time-series classification task and in Section 3.2 for pairwise learning. We would like to add further terminologies in the related work section for a better understanding of non-experienced readers. We would be grateful if the reviewer could point out which of the tasks are unclear, so that we can revise the manuscript accordingly.
> c) We will introduce further network details in a separate section for a better overview of implementation details.
>
> 3. Our approach uses images as training data due to the same reason other methods with cross-modal retrieval do this. These methods work by making use of the information in both (or all) modalities and thereby are able to construct a better latent space. Similarly, in our setup, the image modality helps the time-series modality to learn a better latent representation.
> a) We now make this more clear by adding the following caption: “Comparison of the naive method (left) and our proposed approach (right), where our method shows a much better behaved embedding space compared to the naive approach by learning a joint representation.
> b) Prior work (Ott et al., 2022c) evaluated data augmentation techniques for multivariate time-series (i.e., time warping, scaling, jittering, magnitude warping, and shifting). This approach was rather limited with only 2-3% points of improvement. We will clarify this in Section 5.2.
> c) Figure 9a shows the learned embeddings for the time-series modality without fine-tuning. We realized that the learned representation generalizes well, but misclassifications (e.g., of small and capital letters at the beginning of a word, which happen quite often) also introduce errors in the latent representation. Hence, we analyzed the embeddings of the cross-modal setup in Figure 9b with t-SNE (we expect similar findings with a linear-probing or kNN analysis). The figure shows a better representation of the data with the embeddings of the time-series modality being close to the embeddings of the image modality, and hence, more distinctive clusters with better separation.
>
> 4. The reviewer raises an interesting point when asking for a comparison of the proposed model with existing work such as the one behind the iPad OS. The methods used by iPad OS and others require a tablet with a touch screen surface and stylus pens with integrated magnetometers. These methods can easily reconstruct the trajectory of the pen tip through the magnetometer on the surface, and hence, can classify the written text. This is more challenging when using sensor-enhanced pens, as the classification task is performed directly on the sensor data. One drawback of methods used in the iPad OS is the requirement for writing on specific surfaces, which in turn can influence the writing style. Also, certain applications require writing on normal paper, or the availability of a touch screen surface is not always given, e.g., when writing a short list, but notes need to be digitized afterwards. We will address the differences in our paper now more prominently in Section 1.

---

> ### Author Response · Authors · 2022-10-25
> **Response to Broader Impact Concerns by Reviewer DLNN**
>
> We thank the reviewer for highlighting this particularly important aspect. Machine learning models face various challenges when classifying text with this sensor-enhanced pen. These challenges can appear if there is a domain shift between training and test datasets, e.g., specific writers have a unique writing style and accelerations, or they hold the pen differently. Alsosome writers might have a unique writing environment (different writing surfaces such as a unique table or paper which leads to different magnetic fields). Another difficulty can appear through an under-represented group such as left-handed writers (see Section 5.2, “transfer learning on left-handed writers”) or a disabled person for which the model is not trained on. A well-generalized model trained on all possible pen movements is very challenging and requires a lot of training data. One solution is to record data for a unique writer and adapt the model, or augment the data for a better representation, e.g., as proposed with our method on left-handed writers evaluated in Section 5.2. Hence, unique writers are not excluded and  the task for classifying writing from under-represented groups is addressed in our paper. Domain shifts still remain a challenging problem and we will again emphasize this also with respect to disabilities in our latest revision.

---

### Review · Reviewer_L4if · 2022-11-02

**Summary Of Contributions:**

This paper proposes to extend the triplet loss to cross-modal examples with different labels. Experiments are conducted on handwriting recognition (with multi-sensory data) and handwriting tasks with synthetic (generated) data. The paper proposes a multi-task training setup with the triplet loss over image data against time-series data. They show that this multi-task training setup improves performance on the time-series classification task.


**Audience:**

Yes

**Broader Impact Concerns:**

I have no concerns about ethical implications or broader impact for this paper. It is about handwriting recognition which is not a particularly sensitive topic.

**Claims And Evidence:**

No

**Requested Changes:**

My main concern is with the overall presentation of the paper, which makes it hard to distill the key results. I think the problem setup and experiments are interesting, but it’s difficult for me to understand the primary contributions or experimental findings.

It would be ideal if parts of the manuscript can be re-written (see Weaknesses section) to be more digestible, at present it is difficult to understand. This is especially for someone not already working in the online handwriting recognition area, or for those working on the topic but are unfamiliar with deep metric learning. The overall paper can be tightened to improve the story - at present there is a lot of content presented (on deep metric learning, on handwriting recognition, on modeling), and it is difficult to understand the primary contributions of the paper.

The modeling results need to be clarified: please highlight the overall performance of the proposed model, and contrast it with existing state of the art. It would be best if performance can be explained (if it is worse, why?).


**Strengths And Weaknesses:**

## Strengths:
- The proposed method allows one to make full use of both dense time-series data and sparse image data, which efficiently leverages the full data pipeline. It is encouraging to see work that fuses disparate modalities to improve upon a unimodal model.
- The proposed setting is interesting: combining time-series sensory data from a sensor-enhanced pen with images of handwritten characters in an effort to improve results on the time-series problem.

## Weaknesses:

### Problem setting
The setting of the problem is a little limited in scope, and seems expensive to collect. Data comes from a sensor-enhanced pen (consisting of accelerometers, gyroscope, magnetometer, and force sensors). This application is not my area of expertise, and I am not sure if having access to this sensory data is a reasonable setting for the problem.

### The evaluation is not very convincing.
- The WER and CER on IAM-OffDB is significantly worse for the proposed model compared to existing models. For example, the reported WER on IAM-OffDB is 23.61 (WER) for ScrabbleGAN and 4.70 (CER) for OrigamiNet, while the proposed method seems to be 89.37? (I might be misreading Table 5 here, but I could not identify the reported result for the proposed model on these two datasets).
- It also seems like it is necessary to run ScrabbleGAN and OrigamiNet on the OffHW-German dataset to retrieve baseline results, as the proposed model has a significantly lower WER on this dataset than IAM-OffDB (suggesting that OffHW-German is an easier task).
- Why is the reimplementation of OrigamiNet so much worse on IAM-OffDB CER compared to the reported result? If this is because of less GTR used, it would be useful to run the same baseline model to get scores on OffHW-German and IAM-OffDB.


### Writing of the paper can be greatly improved
- Table 2 is overwhelming. It may be more informative to have a concise and more focused table, consisting of methods that are immediately relevant to the proposed method CMR-IS (perhaps scoped to models that tackle handwriting recognition). For example, it is not clear to me how including methods that operate on visual-question answering or video-to-video similarity are beneficial for readers of the paper.
- Section 3.1 is titled “cross-modal representation learning” but its content describes a very specific problem (i.e., multivariate time-series). There exists many other models and tasks that do not fall into this category. This category should be renamed to be more reflective of its actual purpose and content.
- There are many non-standard (and in my opinion, unnecessary) abbreviations and it makes the paper difficult to read. For example: DML, GTR, MTS, GASF, ED (which is only used 4 other times). It would make the paper much more readable if these were not abbreviated.

---

> ### Author Response · Authors · 2022-11-21
> **Response to Reviewer L4if**
>
> We would like to thank the reviewer for the useful comments and efforts towards improving our manuscript. We address the raised questions by the reviewer in the following:
>
> 1. [Strengths] We thank the reviewer for pointing out the strengths of our manuscript, i.e., the interesting approach to combine time-series sensory data with images of handwritten words.
>
> 2. [Problem setting] The reviewer is correct that applications of the sensor-enhanced pen are limited, but we showed that due to the higher number of prototypical developments, the interest in these applications is high. Furthermore, recent publications came up with similar developments that are only prototypical, for example, Singh & Chaturvedi (2023); He et al. (2022); Alemayoh et al. (2022). Hence, there is already a lot of interest and future technical advancements will further boost this. We will add the mentioned publications in Section 2.2 to better motivate our approach. Further note that the pen developed by STABILO is a finished product and can be bought. Data collection and processing is straightforward and allows applications like ours to be easy to implement in real-world.
>
> 3. [Evaluation] The reviewer is correct that the reported CER of 15.67% on the IAM-OffDB datasets of our OrigamiNet with 4 GTR blocks is higher compared to the 4.7% CER reported by OrigamiNet. The reason is that OrigamiNet (Yousef & Bishop, 2020) reported experiments with a model using 12 GTR blocks, which significantly increases the number of parameters and also the training time. With 8 GTR blocks, OrigamiNet reported a CER of 5.6% and with ResNet-74 a CER of 6.1%. Furthermore, OrigamiNet was trained on multi-line methods, which is an easier task as the image of the paragraph does not have to be segmented into lines. OrigamiNet experimented with different lengths of the images between 700 and 1,500 pixels which significantly changed the results (between 30.34% to 43.14% CER for VGG and 7.45% to 8.12% CER for ResNet-26). We experimented with shorter lengths, which is more sensible for the single word classification. While the images for the multi-line task are of approximately similar lengths, the image lengths of the single line task varies strongly, and hence, zero padding has a high influence on the model performance. This problem does not appear for the OffHW-German dataset, as the dataset contains only single words with similar lengths. Due to the long training times of OrigamiNet with 12 GTR modules and as the important aspect of our cross-modal model is to achieve low CERs on the generated OffHW-German dataset, we did not aim to further improve OrigamiNet on the IAM-OffDB dataset. As OrigamiNet with 4 GTR modules showed very low CERs of 0.11% on the OffHW-German dataset, we selected this method for the following model setup.
>
> 4. [Writing] Indeed, Table 1 and Table 2 contain a lot of methods and are overwhelming. In the previous submission, the reviewers requested to add more related work about offline and online handwriting recognition and cross-modal retrieval. As there is currently no survey about online and offline HWR methods, we included many related techniques to provide a complete overview for the interesting reading. Similarly, for cross-modal retrieval, we added methods to compare the use of the modality and pairwise as well as deep metric learning techniques. We think that this overview can be helpful for other techniques – not familiar with online handwriting recognition. We will select the most relevant techniques in the comparison with our proposed method CMR-IS in the revision of our manuscript, and add the detailed table with a broader comparison in the appendix for the interested reader.
>
> 5. [Writing] The reviewer is correct that the title of Section 3.1 is very general. We will rename the section to “Cross-Modal Retrieval for Time-Series and Image Classification”.
>
> 6. [Writing] We will reduce the use of unnecessary abbreviations (i.e., DML, GTR, MTS, GASF, ED) as suggested by the reviewer.
>
> 7. [Model results] We will add further results of InceptionTime to Table 7 to make a comparison with our baseline model CNN+BiLSTM possible.
>
> Further publications:
>
> Singh & Chaturvedi (2023): S. K. Singh and A. Chaturvedi. Leveraging Deep Feature Learning for Wearable Sensors Based Handwritten Character Recognition. In Biomedical Signal Processing and Control, volume 80(1), February 2023. doi: 10.1016/j.bspc.2022.104198.
>
> He et al. (2022): Q. He, Z. Feng, X. Wang, Y. Wu, and J. Wang. A Smart Pen Based on Triboelectric Effects for Handwriting Pattern Tracking and Biometric Identification. In ACS Appl. Mater. Interfaces, volume 14(43), October 2022. doi: 10.1021/acsami.2c13714.
>
> Alemayoh et al. (2022): T. T. Alemayoh, M. Shintani, J. H. Lee, and S. Okamoto. Deep-Learning-Based Character Recognition from Handwriting Motion Data Captured Using IMU and Force Sensors. In MDPI Sensors, volume 22(20), October 2022. doi: 10.3390/s22207840.

---

### Review · Reviewer_cHWV · 2022-11-29

**Summary Of Contributions:**

The authors present an approach for handwritten digit recognition (HWR). They are mostly interested in online HWR, and specifically the case where a sensor-enhanced pen is used, while writing on normal paper (instead of a stylus pen on a touch screen surface).  They propose to learn a joint space and representation between pairs of image and time-series data, ie offline HWR from generated images (OCR) and online HWR from sensor-enhanced pens, by learning a common representation between both modalities. They learn using a triplet loss.

**Audience:**

Yes

**Claims And Evidence:**

Yes

**Requested Changes:**

* Extensive length: I think that the paper text can generally be way more tight and up to the point. I suggest the related work section significantly trimmed and summarized in the main section and then be a complete review in the appendix.

* Table need better, moer descriptive captions (eg basic explanation of metric abbrev used)

* Table 1 is borderline incomprehensible. I ancourage the athors to move the complete table in the appendix and instead create a clear figure or table that supports the important comparisons and obesrvations, ie what we should be keeping from this table.

* Table 2 is also more a suplementary material table, maybe the variant for the main text could be focusing only one the methods that share one component with CMR-IR, split in sections: ie a section listing methods tackling the exact same modalities, a section with methods using contr. triplet loss, section with methods for HWR. Also, I couldnt find where in the text this table is referenced.

* table 3 is not referenced in the text I think, and needs a clearer caption.

* Table 4 caption seems wrong wrt WD/WI (was this for table 5?)

**Strengths And Weaknesses:**

**Strengths:**

S1) to my knowledge it is an interesting idea to learn a joint space between time-series data from sensor-enhanced pen and text images as it could enable learning a stronger representation for both online and offline HWR.

S2) the use of GANs to create time-series+image pair data for Offline HWR is an interesting idea

S3) The second, current version of this paper is improved and clearer. The claims are not exargerated, terminology is more aligned with the ones used in the community, while the related work section is trully extensive.

**Weaknesses:**

W1) the technical novelty of the paper is limited and the fact that the scope of this paper is narrow, ie this is an application paper for online HWR. this is of course not necessarily  a couse of rejection, given that as I mention above the paper does not exargerate its claims

W2) I think that the paper is now at a place where it is too long and hard to navigate. I think that there are no grounds for this paper to be longer than 12 pages, as most of the added text and data are more supplementary than crusial for following and understanding the paper and its contributions. See suggestions below.

I also want to mention here that HWR is not my area so it is hard for me to understand if all related works are mentioned and compared to in the manuscript.

---

> ### Author Response · Authors · 2022-11-30
> **Response to Reviewer cHWV**
>
> We would like to thank the reviewer for the thoughtful comments and efforts towards improving our manuscript. We address the points raised by the reviewer in the following:
>
> 1. [Strengths S1 to S3] We thank the reviewer for appreciating our work to use GANs to learn a common representation between image and time-series modalities.
>
> 2. [Weakness W2, requested changes: extensive length] We agree with the reviewer that the length of the paper could be reduced and that we list an extensive body of related work (see also the comments of the reviewer L4if). We will shorten the related work section for offline handwriting recognition (i.e., Table 1) to summarize only the most important work to help readability, and provide a complete overview in the appendix.
>
> 3. [Requested changes: table captions] We will make sure that all metrics abbreviations in the tables are introduced and explained in the captions.
>
> 4. [Requested changes: Table 2] As also suggested by reviewer L4if, we provide a short summary of the most important, related work in the corresponding section, and refer the interested reader to the appendix for a complete overview of cross-modal methods.
>
> 5. [Requested changes: Table 2+3] We will make sure that all tables and figures are introduced and referenced.
>
> 6. [Requested changes: Table 4+5] The reviewer is correct. We will update the caption of Table 4 and Table 5.

---

### Author Response · Authors · 2022-11-30
**Comment About Revision**

Dear AC and all Reviewers,

We sincerely appreciate the AC’s and all reviewers’ time and insightful comments, which helped a lot in further improving our paper.

We updated the manuscript with all requested changes (marked in blue font). In particular, we better summarize the related work and only list the most important contributions and further improved writing and clarity of our paper.

---

### Decision · Action_Editors · 2023-01-08

**Recommendation:** Reject

**Comment:**

The reviewers noted that the paper presents an interesting approach to handwriting recognition using signals from a stylus pen. However, they unanimously expressed concerns about the paper's narrow focus on this problem and suggested that it may be more suitable for publication in venues that focus on applications.

All the reviewers raised concerns about the lack of novelty in the paper, which is understandable given that it mainly involves the development of an application based on existing techniques. They also expressed concerns about the lack of sufficient evidence to support the claims made in the paper. While the authors addressed some of the concerns regarding the results on the IAM-OffDB and OffHW-German datasets in their rebuttal, they did not address concerns about the absence of comprehensive ablation studies and the lack of insights from the visualizations.

There were also several issues with the writing quality of the paper, such as its length and the confusing figures and tables, which were not adequately addressed in the authors' response to the reviewers.

Based on the issues mentioned above, we recommend rejecting the submission at this time.

**Audience:**

The reviewers noted that this paper may be more suitable for publication in other venues that focus on applications of AI/ML, as it specifically discusses the use of a particular set of sensors in a specific application (handwriting recognition using a stylus pen equipped with accelerometers, gyroscope, magnetometer, and force sensors), which may not be of broad interest to the TMLR audience.

**Claims And Evidence:**

The reviewers expressed concerns that the paper does not provide sufficient evidence, both quantitative and qualitative, to back up its claims. Even after the authors responded to the reviewers' comments, one reviewer still found the experimental comparisons and visualizations to be inadequate.